# FGL2 promotes tumor progression in the CNS by suppressing CD103⁺ dendritic cell differentiation

Jun Yan[1,2,3], Qingnan Zhao[3], Konrad Gabrusiewicz[4], Ling-Yuan Kong[4], Xueqing Xia[3], Jian Wang[5], Martina Ott[4], Jingda Xu[6], R. Eric Davis[6], Longfei Huo[3], Ganesh Rao[4], Shao-Cong Sun [7], Stephanie S. Watowich [7], Amy B. Heimberger[4] & Shulin Li[3]

Few studies implicate immunoregulatory gene expression in tumor cells in arbitrating brain tumor progression. Here we show that fibrinogen-like protein 2 (FGL2) is highly expressed in glioma stem cells and primary glioblastoma (GBM) cells. FGL2 knockout in tumor cells did not affect tumor-cell proliferation in vitro or tumor progression in immunodeficient mice but completely impaired GBM progression in immune-competent mice. This impairment was reversed in mice with a defect in dendritic cells (DCs) or CD103⁺ DC differentiation in the brain and in tumor-draining lymph nodes. The presence of FGL2 in tumor cells inhibited granulocyte-macrophage colony-stimulating factor (GM-CSF)-induced CD103⁺ DC differentiation by suppressing NF-κB, STAT1/5, and p38 activation. These findings are relevant to GBM patients because a low level of *FGL2* expression with concurrent high *GM-CSF* expression is associated with higher *CD8B* expression and longer survival. These data provide a rationale for therapeutic inhibition of FGL2 in brain tumors.

[1] Center for Brain Disorders Research, Capital Medical University, Beijing 100069, China. [2] Beijing Institute for Brain Disorders, Beijing 100069, China. [3] Department of Pediatrics–Research, The University of Texas MD Anderson Cancer Center, Houston, TX 77030, USA. [4] Department of Neurosurgery, The University of Texas MD Anderson Cancer Center, Houston, TX 77030, USA. [5] Department of Biostatistics, The University of Texas MD Anderson Cancer Center, Houston, TX 77030, USA. [6] Department of Lymphoma and Myeloma, The University of Texas MD Anderson Cancer Center, Houston, TX 77030, USA. [7] Department of Immunology, The University of Texas MD Anderson Cancer Center, Houston, TX 77030, USA. Correspondence and requests for materials should be addressed to A.B.H. (email: aheimber@mdanderson.org) or to S.L. (email: sli4@mdanderson.org)

**B**rain tumor progression, relapse, and invasion are driven by abnormal transcriptional profiles resulting from either intrinsic genetic or epigenetic changes[1,2]. The impact of intrinsic immune-associated genes on brain tumor progression in the presence of a host immune system is much less well understood. Thus far, only the gene for indoleamine 2,3-dioxygenase has been shown to have a role in glioma progression[3]. Over 80% of mice implanted with GL261 gliomas in which this gene was knocked out had long-term survival that was associated with decreased T-regulatory cell (Treg) recruitment by tumors and enhanced T cell-mediated tumor rejection[3]. This result suggests that immune regulatory genes within tumor cells may be the arbitrators of tumor-cell fate in the central nervous system (CNS).

Antigen-presenting cells (APCs) are essential for the induction of adaptive T cell responses[4]. Tumor-associated dendritic cells (DCs) take up, process, and transport tumor antigens to draining lymph nodes for priming and activation of T cells[4]. The transcriptional programs within DCs can influence their immunological role. Batf3-dependent CD103+/CD8a+ DCs are essential for inducing effector T cell recruitment to the tumor and priming T cells in tumor-draining lymph nodes (TDLNs)[5]. It is unknown whether Batf3-dependent DCs have a role in CNS tumors.

Fibrinogen-like protein 2 (FGL2) is a membrane-bound or secreted protein expressed by macrophages, T cells, and tumor cells that has coagulation activity or immune-suppressive functions[6–10]. FGL2 promotes mammary tumor progression by promoting tumor angiogenesis or inducing epithelial-to-mesenchymal transition[10]. We previously showed, using an engineered gene expression system in mouse glioma cells, that FGL2 is a key hub of tumor-mediated immune suppression in glioblastoma multiforme (GBM) by regulating expression of immune checkpoints and augmenting intratumoral skewing of Tregs, myeloid-derived suppressor cells (MDSCs), and M2 cells[8]. However, the exact functional role of FGL2 at both the molecular and cellular levels remains largely unknown. Likewise, the connection between FGL2 and CD103+ DCs is totally unknown. To determine the effect of tumor-cell intrinsic FGL2 on tumor progression, we used complete FGL2 knockout (KO) tumor-cell lines and FGL2-deficient ($FGL2^{-/-}$) mice to reveal a novel molecular immune mechanism for FGL2's regulation of tumor progression in the CNS. We discovered that FGL2 secreted from tumor cells is the primary suppressor of differentiation of antigen-presenting CD103+ DCs via blocking of granulocyte-macrophage colony-stimulating factor (GM-CSF) signal transduction and subsequent T cell functions in the brain. By contrast, an FGL2KO host ($FGL2^{-/-}$) has no impact on inhibition of FGL2-positive tumors. Cumulatively, the data indicate that tumor cell-expressed FGL2 serves as a critical onco-immune target.

## Results

**Heterogeneous expression of FGL2 in GBM.** FGL2 is primarily expressed in immune cells, including T cells, macrophages, and natural killer (NK) cells[6,9]. Our previous data show that FGL2 is highly expressed in glioma and is associated with poor prognosis in glioma patients[8]. To further characterize the cellular sources of FGL2 expression in GBMs, quantitative western blotting was performed among different cell types, including normal cells and glioma stem cells (GSCs). FGL2 was minimally expressed in normal neuron cells (HCN-1A), endothelial cells (HMVEC-L), and primary human peripheral blood monocytes (PBMCs) from healthy donors relative to the GSCs (GSC7-2, GSC11, GSC20, and GSC28) (Fig. 1a). Flow cytometric analysis of FGL2 expression showed 1.7–11.5% FGL2 positive-staining glioma cells among disaggregated GSC neurospheres (Fig. 1b; Supplementary Figure 1). Single-cell suspensions derived from fresh ex vivo

GBM specimens demonstrated that FGL2 expression occurred primarily in the glioma subpopulation that expressed glia-specific marker GFAP (43.3 ± 14.3% of GFAP+ cells). Portions of CD45+ immune cells (37.3 ± 18.3% of CD45+ cells) and CD31+ endothelial cells (47.7 ± 14.7% of CD31+ cells) were also FGL2 positive, but endothelial cells (1.0 ± 0.6%) and leukocytes (5.7 ± 3.0%) accounted for only small fractions of total cells from these tissues (Fig. 1c). Immunohistochemical analysis of GBM specimens demonstrated abundant co-localization of FGL2 expression with brain tumor cells using the GFAP marker (15.8 ± 4.3%), moderate co-localization with immune cells (based on CD45 co-expression [6.5 ± 1.6%]), and occasional association with endothelial cells (denoted as CD31+ [3.3 ± 1.4%]) (Fig. 1d), consistent with the flow cytometry data. Cumulatively, these data indicate that the predominant cellular sources of FGL2 within the GBM are the tumor cells and GSCs.

**Intrinsic FGL2 in tumor cells promotes tumor progression.** Given the heterogeneous but typically high levels of FGL2 expression in GBM cells, we next wanted to analyze the specific role of FGL2 within tumor cells in brain tumor progression. Stable FGL2KO tumor-cell lines were generated utilizing CRISPR/Cas9 technology. Deletion of the DNA fragment in FGL2 exon 1 in each clone was confirmed by gene sequencing (Fig. 2a). Western blotting analysis showed complete knockout of FGL2 expression in glioma (GL261-FGL2KO) and Lewis lung cancer cells (LLC-FGL2KO) and downregulated FGL2 expression in FGL2-knockdown astrocytoma cells (DBT-FGL2KD) compared with the cognate cells in which the non-targeting control sequence was introduced via the same technology (Fig. 2b). As expected, mice bearing tumors generated by implantation of control GL261 tumor cells (GL261-Ctrl) experienced rapid disease progression and died. In contrast, immune competent mice, including wild-type (WT) and $FGL2^{-/-}$, survived after implantation of GL261-FGL2KO tumor cells (Fig. 2c, d). Similar results were obtained in mice implanted with a high (5-fold) or a maximal (20-fold) number of GL261-FGL2KO tumor cells (Supplementary Figure 2a-d). LLC was selected for this experiment because lung cancers are the most common source of brain metastasis, with 30~60% of lung tumor patients developing brain metastasis, a major cause of death[11,12]. Like GL261-FGL2KO tumor cells, LLC-FGL2KO and DBT-FGL2KD tumor cells showed no progression in immunocompetent mice (Fig. 2e, f). Therefore, FGL2 expression in the tumor cell but not in the host is required for tumor progression in immune-competent mice (Supplementary Figure 2e). Notably, this was not secondary to FGL2's impact on the cell growth rate, because both FGL2KO and Ctrl glioma cells proliferated equally in vitro (Supplementary Figure 2f). The tumor cell-specific FGL2KO-mediated impairment of tumor progression was also not due to differences in implantation, because bioluminescence imaging showed similar chemoluminescence signals in both the Ctrl (3.44E + 07) and FGL2KO (3.90E + 07) tumors on day 1. The luciferase signal was rapidly reduced in the FGL2KO tumor cell-implanted mice (5.149e + 06) while increasing markedly in Ctrl tumor cell-implanted mice (9.52e + 09) by day 7, illustrating the marked difference in tumor progression between Ctrl and FGL2KO tumor-bearing mice (Fig. 2g).

**FGL2KO-mediated tumor suppression depends on CD8+ T cells.** To understand the mechanism of FGL2KO tumor cell-mediated inhibition of tumor progression in mice, the progression of GL261-Ctrl and GL261-FGL2KO tumors was studied in immune-deficient mice. In NOD-scid gamma (NSG) mice, which are deficient in T, B, and NK cell functions, implantation of Ctrl

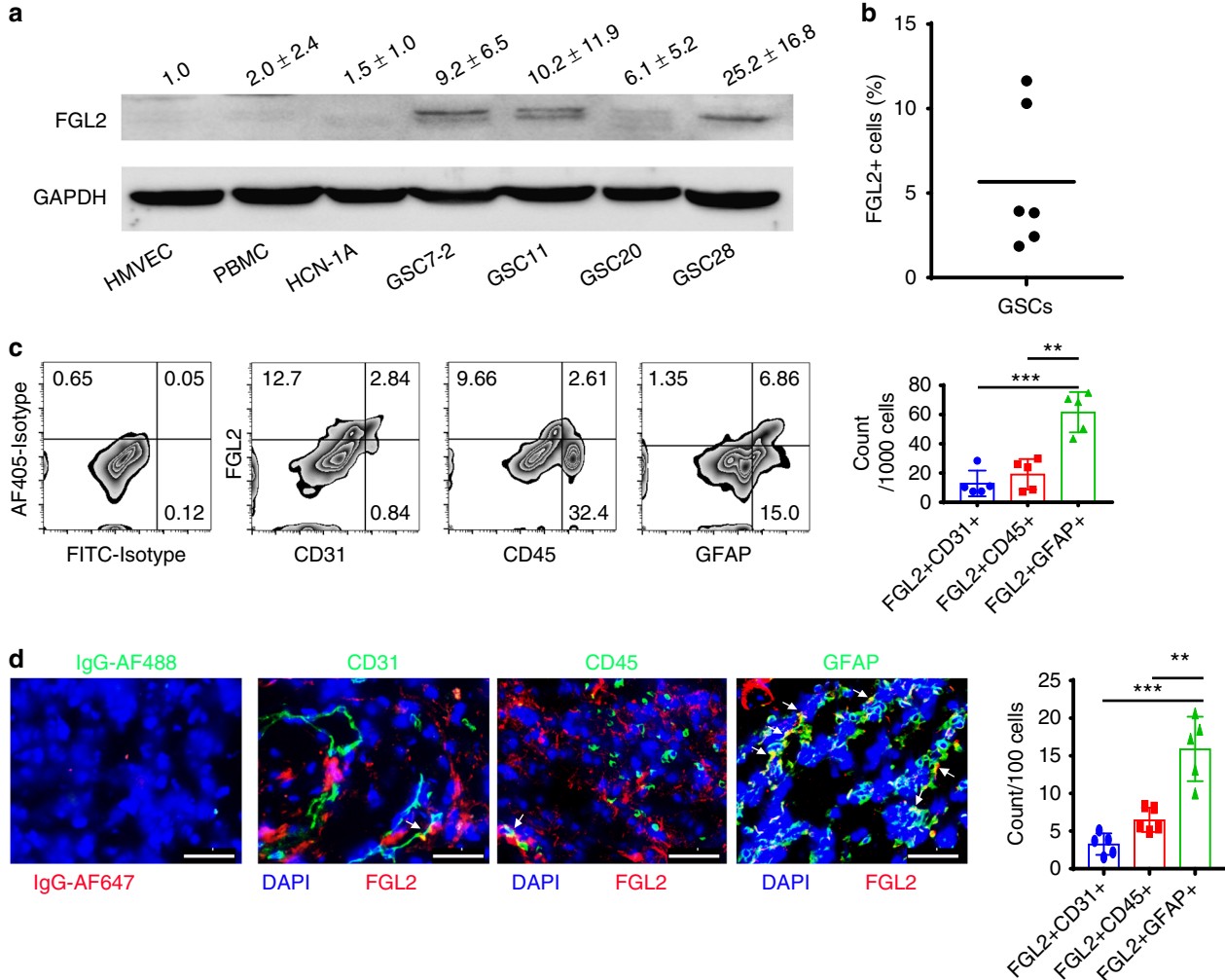

**Fig. 1** FGL2 is expressed by glioma cells. **a** FGL2 protein levels were detected by western blot in human peripheral blood monocytes (PBMCs) from healthy donors, normal endothelial cells (HMVECs), normal neurons (HCN-1A), and four glioma stem cell lines (GSC7-2, GSC11, GSC20, and GSC28). Western blots shown are representative, and quantities are summarized as ratio changes from three independent experiments. Glyceraldehyde 3-phosphate dehydrogenase (GAPDH) was used as a protein-loading control. **b** Percentage of FGL2$^+$ cells in cultured human GSCs. **c** Representative flow cytometry plots of FGL2 expression on CD45$^+$, CD31$^+$, or GFAP$^+$ cells from tumors of patients with newly diagnosed GBM. Data ($n = 5$) are presented as the mean ± S.D. and analyzed by one-way ANOVA. **d** Sections of clinical GBM samples were double stained for co-expression of FGL2 (red) and GFAP (green), or of FGL2 (red) and CD45 (green), or of FGL2 (red) and CD31 (green). Nuclei were counterstained with DAPI (blue). White arrows point to co-localization. Scale bars, 50 μm. Data are presented as the mean ± S.D., and representative images are shown ($n = 5$). A one-way ANOVA was used to calculate the two-sided $P$ values. Significant results were presented as **$P < 0.01$, ***$P < 0.001$

or *FGL2KO* tumor cells resulted in equivalent survival times and tumor progression-associated death (Fig. 3a), indicating that the immune system is required for tumor cell-specific *FGL2KO*-mediated tumor regression. To clarify which immune cell populations participate in the regression of *FGL2KO* tumors, we used immune cell depletion and T cell-deficient mice. Depletion of CD4$^+$ T cells had no effect on the survival of mice implanted with *FGL2KO* tumors (Fig. 3b; Supplementary Figure 3). Whereas NK depletion had a modest negative effect on survival ($P = 0.14$, Fig. 3c), depletion of CD8$^+$ T cells completely reversed the survival benefit to mice implanted with GL261-*FGL2KO* cells (Fig. 3d; Supplementary Figure 3). Consistent with the results of CD8$^+$ T cell depletion, tumor progression was re-established in 70–80% of mice with a *CD8$^{-/-}$* background implanted with either GL261-*FGL2KO* or LLC-*FGL2KO* tumor cells (Fig. 3e–g). The number of CD8$^+$ T and CD4$^+$ T cells significantly greater in the brains of mice implanted with *FGL2KO* tumor cells than in those implanted with *Ctrl* tumor

cells, but this was not the case with NK or NKT cells (Supplementary Figure 4c). Furthermore, lymphocytes from tumor-draining lymph nodes (TDLNs) and spleens of *FGL2KO* tumor-bearing mice showed higher cytolytic activity against tumor cells than those from Ctrl tumor-bearing mice, indicating a heightened cytotoxic function of CD8$^+$ T cells in mice with *FGL2*-deficient tumors (Fig. 3h). These data strongly support the idea that CD8$^+$ T cells were responsible for suppression of *FGL2KO* tumors.

The increased cytolytic activity of cells derived from TDLNs and spleens after implantation of *FGL2KO* tumor cells suggests the induction of systemic protective adaptive immune responses (Fig. 3h). To validate this assumption, we rechallenged mice with GL261-*Ctrl* cells 45 days after *FGL2KO* tumor implantation to test the presence or absence of CD8$^+$ T cell-dependent recall responses. All mice survived tumor rechallenge (Fig. 3i). The same results were obtained in *FGL2$^{-/-}$* mice prechallenged with GL261-*FGL2KO* cells (Fig. 3j), showing that tumor cell-specific

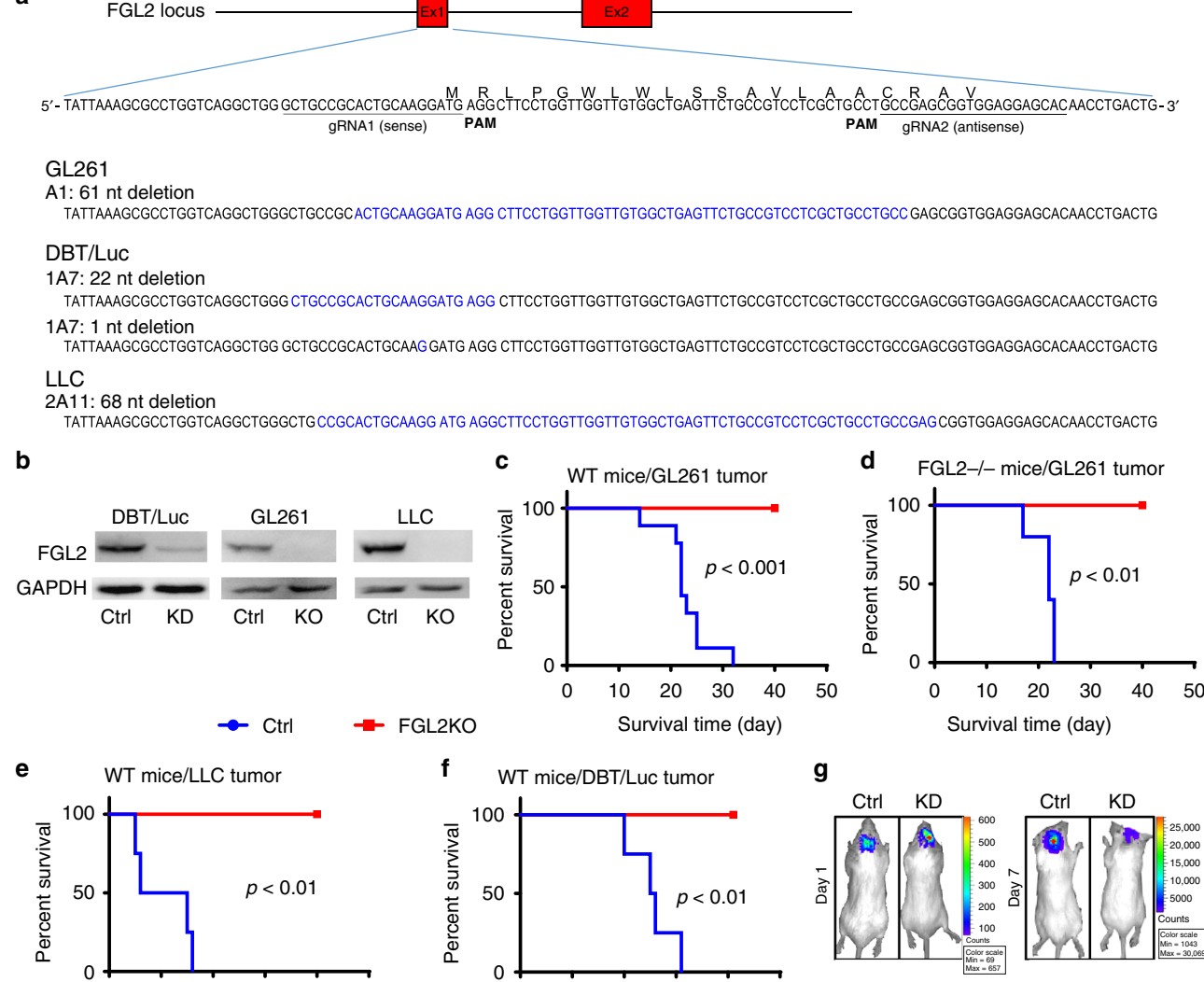

**Fig. 2** FGL2 knockout in tumor cells abolishes tumor progression. **a** Results of DNA fragment deletion in FGL2 exon 1 in individual clones were validated by gene sequencing. Paired gRNAs were designed to excise exon 1 at the mouse FGL2 locus. Individual clones isolated from cells transfected with gRNAs were assayed for deletions and inversions by RT-PCR. **b** Expression level of FGL2 in three tumor-cell lines, control (Ctrl) and FGL2-knockout (KO) or knockdown (KD) tumor cells, was detected by western blotting. The western blots shown represent three independent experiments. **c** Survival curve of wild-type (WT) immunocompetent C57BL/6 mice implanted with GL261-*Ctrl* or GL261-*FGL2KO* tumor cells ($5 \times 10^4$ cells per mouse; $n = 9$ per group). **d** Survival curve of *FGL2$^{-/-}$* mice implanted with GL261-*Ctrl* or GL261-*FGL2KO* tumor cells ($5 \times 10^4$ cells per mouse; $n = 5$/group). **e** Survival curve of WT immunocompetent C57BL/6 mice implanted with Lewis lung cancer (LLC)-*Ctrl* or LLC-*FGL2KO* tumor cells ($5 \times 10^4$ cells per mouse; $n = 4$ per group). **f** Survival curve of WT immunocompetent BALB/C mice implanted with mouse astrocytoma (DBT)-*Ctrl* cells or DBT-*FGL2KD* tumor cells ($5 \times 10^4$ cells per mouse; $n = 4$ per group). **g** Bioluminescence imaging showing DBT-*Ctrl* and DBT-*FGL2KD* tumors at day 1 and day 7 after tumor-cell implantation. All data are representative of at least two independent experiments. The survival curves were analyzed by Kaplan–Meier analysis and the log-rank test was used to compare overall survival between groups

*FGL2KO* is crucial to induce this systemic adaptive immune response. We rechallenged the same mice with GL261-*Ctrl* cells a second and third time, with a gap of 45 days between each challenge. At the second rechallenge, no mice developed tumors, but on the third rechallenge, 5 of 7 mice developed a tumor (Fig. 3i). To ensure that the GL261-*Ctrl* cells used in the challenges were tumorigenic, the same cells were injected into naive WT mice and, as expected, these mice developed tumors (Fig. 3i, j). These results collectively suggest that depletion of *FGL2* from tumor cells induced CD8$^+$ T effector cells and adaptive immune memory against the parental *Ctrl* tumor cells (GL261-*Ctrl*).

**CD103$^+$ DCs accumulation augmenting CD8$^+$ T cell response.** Several steps are necessary to generate a systemic adaptive anti-tumor immune response that leads to effective tumor elimination. First, antigens released by tumor cells are taken up and processed by APCs to present to T cells, inducing T cell priming and activation; second, activated T cells must traffic to the tumor and infiltrate; last, tumor cells are recognized and killed by the antigen-specific T cells[13]. We found no difference in the migration and survival of activated CD8$^+$ T cells treated with conditioned medium (CM) from either *FGL2KO* tumor cells (*FGL2KO*-CM) or *Ctrl* tumor cells (*Ctrl*-CM) in vitro (Supplementary Figure 5a, 5b). CM was used in vitro because FGL2 is a secreted

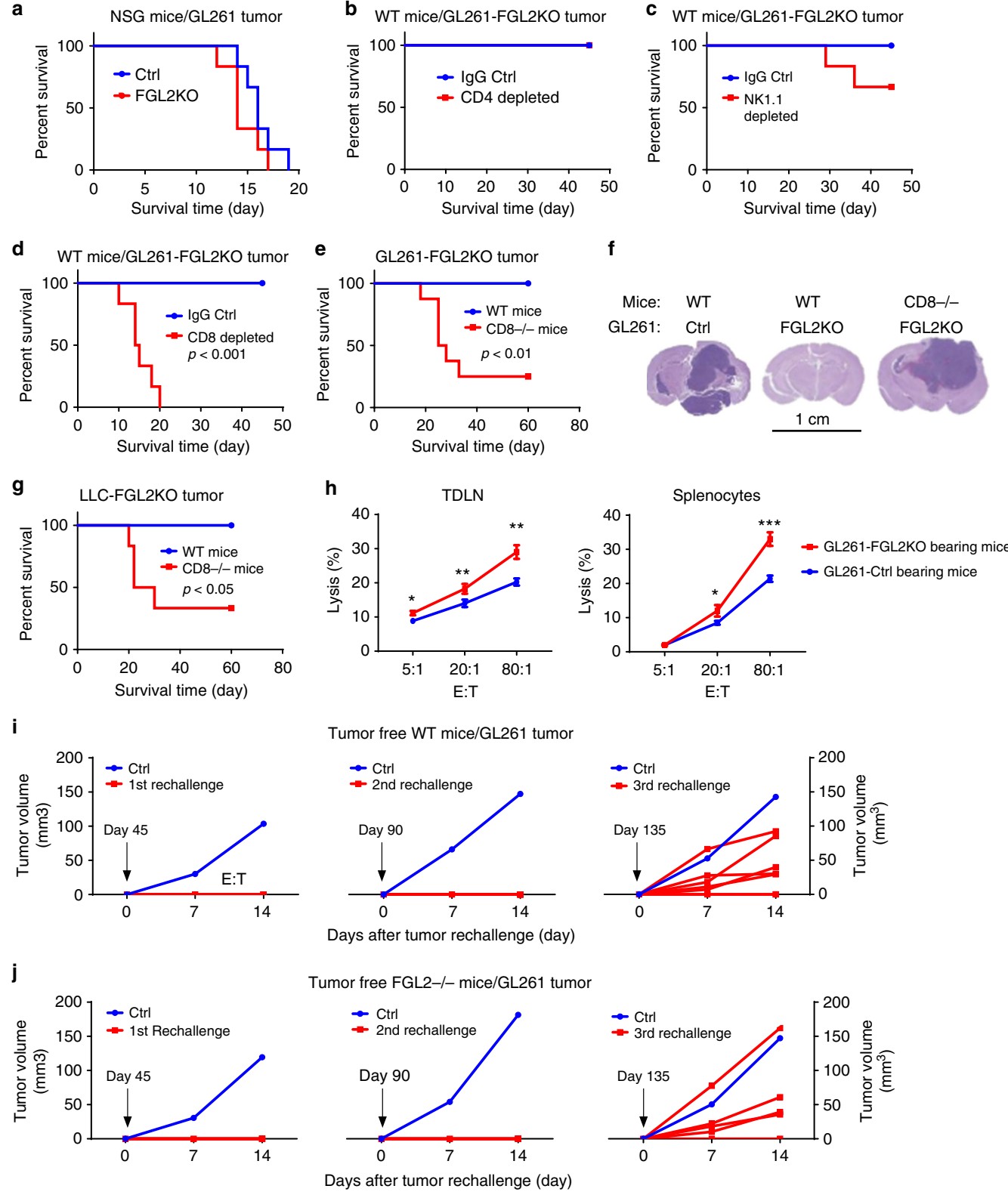

protein and the secreted FGL2 is crucial to causing immune suppression[14]. However, this secretion has no immediate impact on T cell killing activity, since both ovalbumin (OVA)-expressing *FGL2KO* and control tumor cells had the same sensitivity to cell killing by OVA peptide-activated CD8+ T cells (OT-I) in vitro (Supplementary Figure 5c). Instead, this secreted FGL2 suppressed T cell expansion (CFSE dilution), but not activation

(CD25 expression) in vitro (Supplementary Figure 5d). These results suggest that an initial step in the antitumor immune responses, such as antigen presentation, is affected by *FGL2KO* in tumor cells, which ultimately affects the capacity to trigger the CD8+ T cell antitumor immune response.

To successfully trigger the initial steps of priming/activation and expansion of functional T cells, multiple signals must be

**Fig. 3** CD8$^+$ T cell-dependent clearance of *FGL2KO* tumor. **a** Survival analysis of immunodeficient NSG mice implanted with GL261-*Ctrl* or GL261-*FGL2KO* tumor cells ($5 \times 10^4$ cells per mouse; $n = 6$ per group). **b–d** Survival analysis of WT immunocompetent C57BL/6 mice implanted with GL261-*FGL2KO* tumor cells and treated with anti-CD4 (GK1.5), anti-NK1.1 (PK136), or anti-CD8 (2.43) depletion antibody or IgG as control (Ctrl) ($n = 6$ per group). **e** Survival analysis of WT mice and *CD8$^{-/-}$* mice implanted with GL261-*FGL2KO* tumor cells ($n = 6$–8 per group). **f** H&E-stained histopathological sections of brain tissues from representative mice from each group shown in **e**. Scale bar = 1 cm. **g** Survival analysis of WT mice and CD8$^{-/-}$ mice implanted with LLC-*FGL2KO* tumor cells ($n = 6$ per group). **h** Cytolytic activity of CD8$^+$ T cells from tumor-bearing mice against GL261 tumor cells. Lymphocytes from tumor-draining lymph nodes (TDLNs) or spleens of mice implanted with *FGL2KO* tumor cells caused significantly more tumor lysis than those from mice implanted with Ctrl tumor cells ($n = 3$ mice per group). Two-way ANOVA was used to calculate the two-sided $P$ values. E:T, the ratio of effector cells to target cells. *$P < 0.05$, **$P < 0.01$, ***$P < 0.001$ compared with GL261-*Ctrl* tumor-bearing mice. **i** Tumor rechallenge by subcutaneous implantation of GL261-*Ctrl* cells at days 45, 90, and 135 days after primary challenge of GL261-*FGL2KO* by intracerebral implantation in WT mice ($n = 7$). Subcutaneous growth of GL261-*Ctrl* tumors in age-matched naive WT mice challenged with GL261-*Ctrl* cells at the same times but without primary challenge is also shown. **j** The experiment shown in **i** was repeated in *FGL2$^{-/-}$* mice ($n = 6$). All data are representative of at least two independent experiments. The survival curves were analyzed by Kaplan–Meier analysis and the log-rank test was used to compare overall survival between groups

carefully orchestrated by the APCs[15]. Recent publications have shown that the Batf3-lineage CD103$^+$/CD8a$^+$ DCs are the most effective APCs in cross-presentation of tumor antigens to CD8$^+$ T cells for peripheral tumors[16,17]. Notably, the population of Batf3-dependent cross-presenting DCs—migratory CD103$^+$ DCs (CD45$^+$CD11b$^-$MHCII$^+$Lin$^-$CD11c$^+$CD103$^+$)—was greater in the brains and TDLNs of mice implanted with *FGL2KO* tumor cells than in those of mice implanted with Ctrl tumor-cells (Fig. 4a), and the population of CD8a$^+$ DCs (CD45$^+$CD11b$^-$MHCII$^+$Lin$^-$CD11c$^+$CD8a$^+$) was not significantly different between the two groups, showing that numbers of migratory antigen-presenting DCs was increased by intracerebral implantation of *FGL2KO* tumor cells. Furthermore, *Batf3*-deficient (*Batf3$^{-/-}$*) mice implanted with *FGL2KO* tumor cells lacked both CD103$^+$ DCs and CD8a$^+$ DCs in their brains and TDLNs compared with WT C57BL/6 mice (Fig. 4a), suggesting that *FGL2KO* tumor cell-mediated CD103$^+$ DCs development depends on *Batf3*, which is known as a crucial regulator of CD103$^+$/CD8a$^+$ DCs in other tumor model systems[18–20]. We then asked whether DCs, especially CD103$^+$ DCs, controlled the antitumor effect induced by *FGL2KO*. *CD11c-DTR* mice were used to investigate the interaction between DCs and T cells. DT toxin, injected once, resulted in rapid ablation of DCs and complete restoration of DCs at day 6 (Supplementary Figure 6a). As expected, *FGL2KO* tumors progressed when DCs were depleted, and regressed after DCs were restored (Fig. 4b). To further confirm that Batf3-linage CD103$^+$ DCs are required for the antitumor response (for an effector T cell response against tumors), *FGL2KO* tumors were implanted in *Batf3$^{-/-}$* mice (Fig. 4c). As expected, *FGL2KO* tumors were not eradicated in these mice as they were in the WT mice, showing that CD103$^+$ DCs are required for inducing an effective CD8$^+$ T cell response against *FGL2KO* tumor cells. Together, these results suggest that *FGL2KO* tumor cells are more potent than the parental tumor cells at inducing development of Batf3-lineage DCs, which are necessary for initiating an antitumor immune response.

To investigate the capacity of DCs to present endogenously loaded antigens, DCs were isolated from OVA-tumor-bearing mice for direct ex vivo analysis of antigen presentation. Co-culture of naive OT-I cells with DCs found a greater induction of cross-priming to OT-I cells from DCs of *FGL2KO* tumors than from DCs of *Ctrl*-OVA tumors (Fig. 4d). To determine the in vivo antigen-presentation capacity of DCs, exogenous OT-I cells were labeled with CFSE and adoptively transferred into OVA tumor-bearing CD45.1 mice. Mice implanted with *FGL2KO*-OVA tumor cells drew a greater accumulation of OT-I cells in their brains, with an associated higher proliferation of CFSE-labeled OT-I cells in the TDLNs than those of *Ctrl*-OVA tumor-implanted mice (Fig. 4e). This result shows that DCs from *FGL2KO*-OVA tumors have a more efficient antigen-presentation

capacity than those from *Ctrl*-OVA tumors to promote antigen-specific priming and expansion of T cells in the TDLNs and their subsequent accumulation in the brain. To address whether CD8$^+$ T cells are primed more efficiently in the *FGL2KO* tumor-bearing mice than in Ctrl tumor-bearing mice, T-bet expression was measured because it is required for differentiation and function of cytotoxic CD8$^+$ T cell effectors[21]. Greater numbers of CD8$^+$T-bet$^+$ Eomes$^-$ T cells were detected in both the brains and TDLNs at 7 days post implantation in mice implanted with *FGL2KO* tumor cells than in mice implanted with Ctrl tumor cells (Fig. 4f).

TDLNs are the primary sites for cross-priming of CD8$^+$ T cells by CD103$^+$/CD8a$^+$ DCs during tumor growth[22–24]. Immune modulator phosphorylated fingolimod (FTY720) acts as an agonist on sphingosine-1-phosphate receptors in vivo, inducing subsequent internalization of S1P1 and thereby inhibiting DC migration from the periphery to lymph nodes and preventing egress of primed T cells from lymph nodes to target sites[19,25]. Upon administration of FTY720 in *FGL2KO* tumor-bearing WT mice, the tumors progressed (Fig. 4g), suggesting that T cells primed by CD103$^+$ DCs in the TDLNs are required for inducing the adaptive antitumor T cell immune response in mice with *FGL2KO* tumor. *FGL2KO* in tumor cells resulted in increased T cell frequency in tumors and promoted T cell proliferation in TDLNs. These results suggest that T cell priming in the TDLNs is required to promote the antitumor response to *FGL2KO* tumors. Thus, expression of FGL2 in tumor cells prevents T cell priming by reducing the development of Batf3-dependent CD103$^+$ DCs.

**FGL2 suppresses CD103 induction by blocking GM-CSF signaling.** To verify the effect of *FGL2KO* and *Ctrl* tumor cells on generation of CD103$^+$ DCs in vitro, bone marrow cells were cultured with medium conditioned by *FGL2KO* or *Ctrl* tumor cells with/without hFLT3L (50 ng/mL), and CD103$^+$ DC differentiation was measured. In accordance with the in vivo results, the expression level of CD103 was significantly higher on the CD11c$^+$B220$^-$ DCs generated through culture with *FGL2KO*-CM than on those cultured with *Ctrl*-CM on both day 5 and day 15 (Fig. 5a, Supplementary Figure 6b). Clec9A, a specific and universal marker of mouse and human Batf3-dependent DCs which promotes DC cross-priming of cytotoxic T lymphocytes[26,27], was expressed on the cultured DCs at day 15 but not day 5 (Fig. 5a). These data indicate that more CD103$^+$ DCs were differentiated from bone marrow cells in the absence of *FGL2* at day 15 of culture, but CD103 expression on cultured bone marrow DCs was detected as early as day 5. We then examined the capacity of 15 days-cultured bone marrow DCs to stimulate OT-I cells when given OVA protein. Bone marrow DCs

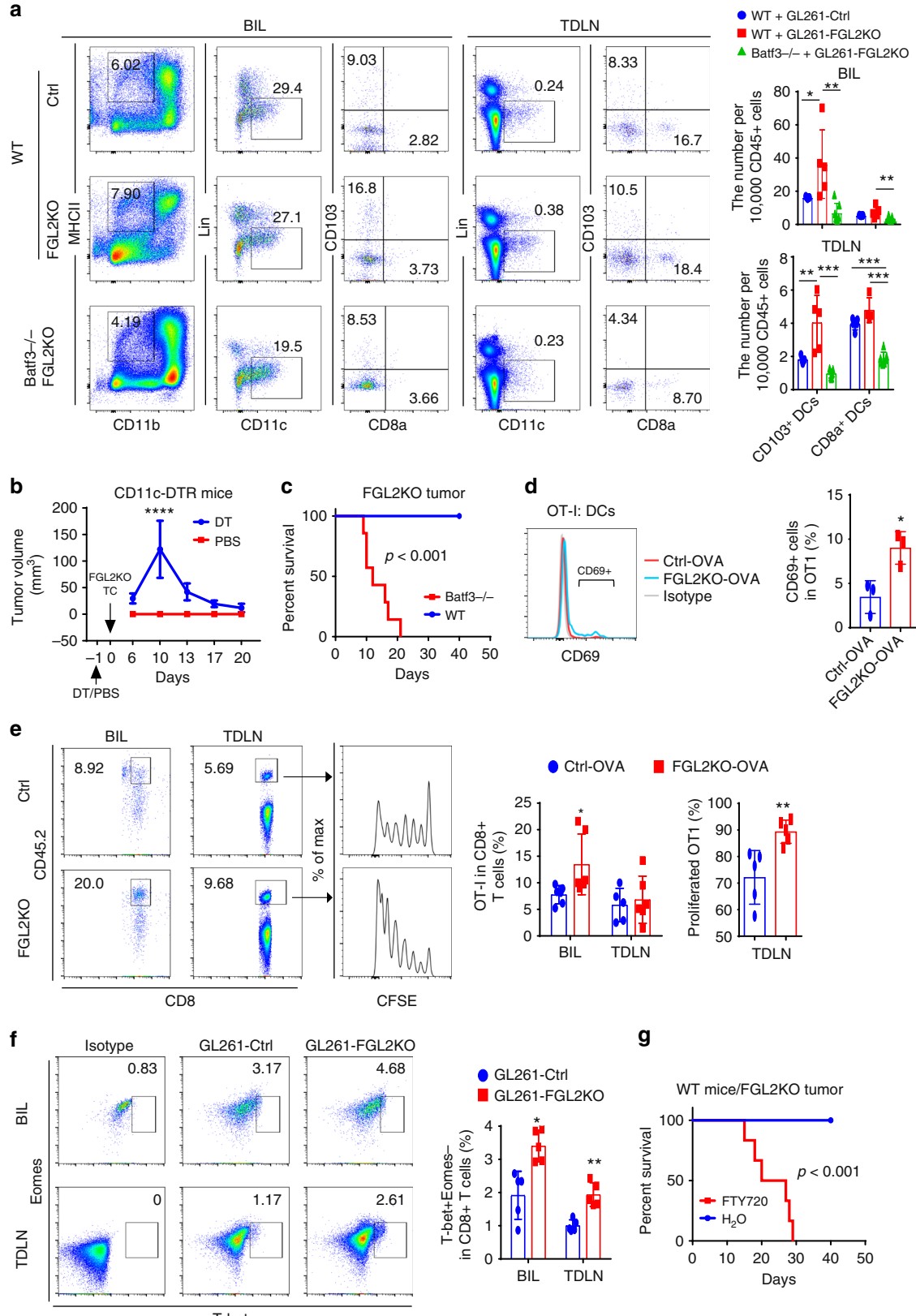

cultured with *FGL2KO*-CM were more efficient than those cultured with *Ctrl*-CM at inducing expression of CD69 on OT-I cells (Fig. 5b), indicating that DCs cultured with *FGL2KO*-CM had greater cross-presentation capacity than DCs cultured with *Ctrl*-CM.

It is well known that FLT3L and GM-CSF promote development of CD103[+] DCs[28,29]. To discern whether either factor was differentially expressed in *FGL2KO*-CM, we evaluated the abundance of FLTL3 and GM-CSF in the CMs. To our surprise, levels of GM-CSF and FLT3L were almost identical in media

**Fig. 4** Batf3-dependent CD103+ dendritic cells are required for CD8+ T cells priming. **a** Brain-infiltrating leukocytes (BIL) and tumor-draining lymph nodes (TDLNs) of Ctrl or FGL2KO tumor-bearing wild-type (WT) mice and *FGL2KO* tumor-bearing *Batf3$^{-/-}$* mice were analyzed for CD103$^+$/CD8a$^+$ dendritic cell (DCs) populations on day 7 post tumor implantation. Data are presented as the mean ± S.D. and were analyzed by one-way ANOVA ($n = 5–7$ per group). **b** *FGL2KO* tumor progression in *CD11c-DTR* mice treated with diphtheria toxin (DT) or PBS. *FGL2KO* tumor cells were implanted subcutaneously into *CD11c-DTR* mice on day 1 after injection of DT or PBS. Data are presented as the mean ± S.D. ($n = 8$ per group). **c** Survival analysis of *Batf3$^{-/-}$* mice implanted with GL261-*FGL2KO* tumor cells ($n = 7$ per group). **d** FACS assay of CD69 on naive OT-I CD8$^+$ T cells co-cultured with DCs isolated from tumors at a ratio of 1:2. Data are presented as the mean ± S.D. ($n = 3$ mice per group) and were analyzed by t-test. **e** Abundance and proliferation of CFSE-labeled OT-I cells in brains and TDLNs of GL261-*Ctrl-OVA* or GL261-*FGL2KO-OVA* tumor-bearing mice. CFSE-labeled OT-I cells were adoptively transferred to tumor-bearing CD45.1 mice on day 2 after tumor implantation. Brains and TDLNs were collected 5 days later for analyzing the presence and proliferation of the CD45.2$^+$CD8$^+$ T cell population. Data are presented as the mean ± S.D. ($n = 5$-6 per group) and were analyzed by t-test. **f** Representative cytometric analysis of CD8$^+$ T cell priming (CD8$^+$T-bet$^+$Eomes$^-$) in BIL and TDLNs at 7 days after tumor implantation. Data are presented as the mean ± S.D. ($n = 5$ per group) and were analyzed by t-test. **g** Survival analysis of WT mice implanted with GL261-*FGL2KO* tumor cells treated with FTY720. Mice were treated with 1 mg/kg FTY720 1 h before tumor-cell implantation and were maintained with drinking water containing 2 μg/mL FTY720 for the duration of the experiment ($n = 6$ per group). All data are representative of at least two independent experiments. The survival curves were analyzed by Kaplan–Meier analysis and the log-rank test was used to compare overall survival between groups. Significant results were presented as *$P < 0.05$, **$P < 0.01$, ***$P < 0.001$

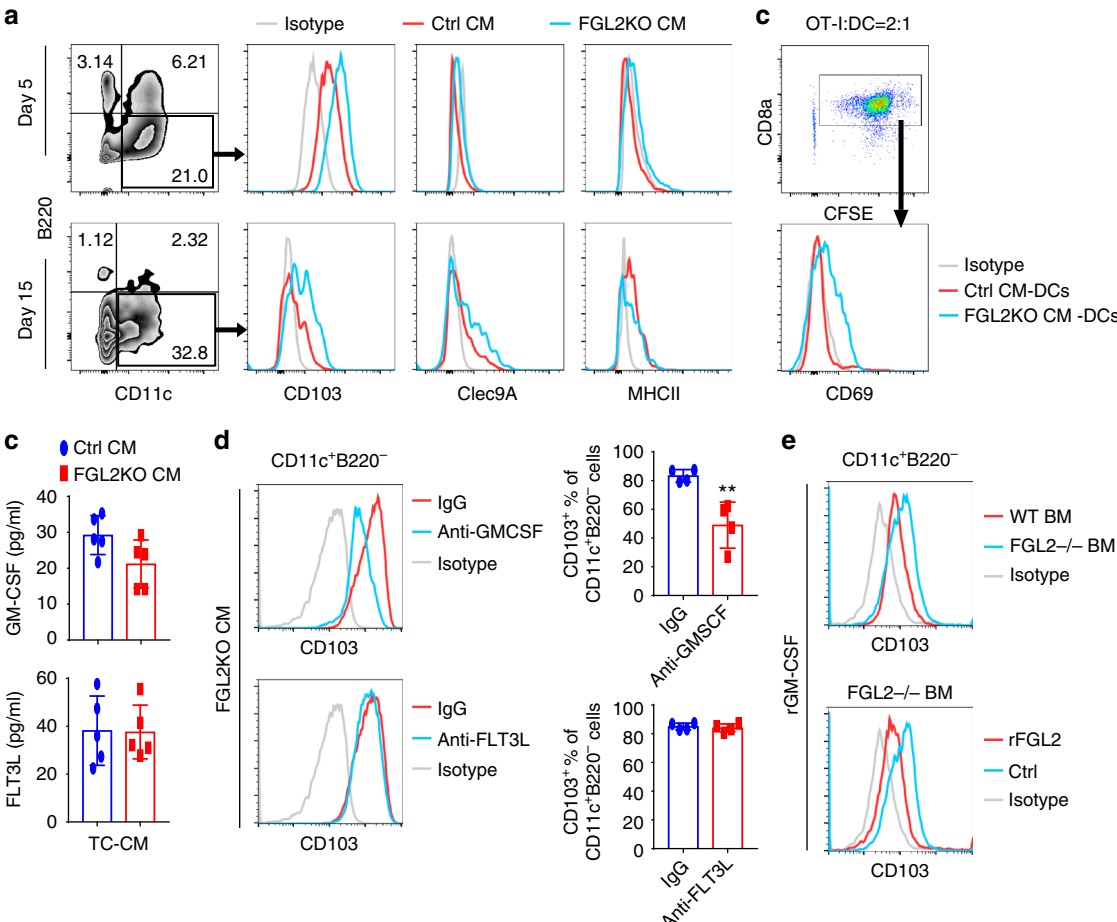

**Fig. 5** FGL2 suppresses GM-CSF-dependent CD103 induction on cDCs in vitro. **a** Expression of CD103 on CD11c$^+$B220$^-$ DCs cultured with conditioned medium (CM) for 5 or 15 days. Bone marrow cells were isolated from C57BL/6 mice and then cultured with *Ctrl-CM* (CM from *Ctrl* tumor cells) or *FGL2KO-CM* (CM from *FGL2KO* tumor cells) for 5 days or 15 days. Non-adherent cells were recovered, stained, and analyzed by FACS to determine the expression levels of CD103, Clec9A, and MHCII on the CD11c$^+$B220$^-$ DCs. **b** OT-I T cell activation by OVA-pulsed CM-cultured bone marrow DCs. Non-adherent bone marrow DCs were recovered from CM-cultured bone marrow DCs at 15 days. The bone marrow DCs ($2 \times 10^4$) were incubated with soluble OVA protein (100 μg/mL) for 2 h and then washed twice before co-culturing with the CFSE-labeled OT-I cells ($5 \times 10^4$). OT-I activation was measured by CD69 expression level, which was determined by flow cytometry. **c** Protein levels of GM-CSF and FLT3L in medium conditioned by tumor cells were detected by ELISA. Data were summarized from five independent experiments and were analyzed by paired t-test. **d** Representative FACS assay of CD103 expression on CD11c$^+$B220$^-$ DCs cultured with FGL2KO-CM in the presence of neutralization antibody anti-GM-CSF (5 μg/mL) or anti-FLT3L (5 μg/mL) for 5 days. Data were summarized from four independent experiments and were analyzed by paired t-test. **$P < 0.01$ comparing with IgG. **e** Representative FACS assay of CD103 expression on wild-type (WT) and *FGL2KO* bone marrow (BM) cells cultured with rmGM-CSF (10 ng/mL) for 7 days with or without rFGL2 (200 ng/mL). All plots are representative of three independent experiments

conditioned by GL261-*FGL2KO* or GL261-*Ctrl* tumor cells (Fig. 5c). While neutralization of GM-CSF in *FGL2KO*-CM markedly decreased CD103 expression on CD11c⁺B220⁻ DCs, however, neutralization of FLT3L had no effect on CD103 expression (Fig. 5d). In this regard, we investigated the effect of *FGL2* deficiency in bone marrow cells on CD103 induction by GM-CSF. Interestingly, CD103 expression was detected on CD11c⁺B220⁻ DCs derived from both WT and *FGL2⁻/⁻* bone marrow cells cultured with recombinant murine GM-CSF (rGM-CSF). However, rGM-CSF induced a higher level of CD103 expression on *FGL2⁻/⁻* DCs than on WT DCs (Fig. 5e). Moreover, the expression of CD103 on *FGL2⁻/⁻* DCs induced by rGM-CSF was suppressed by rFGL2 (Fig. 5e). These data indicate that *FGL2KO* tumor cells promote CD103 expression on DCs by a GM-CSF-dependent mechanism, which is suppressed by FGL2 secreted from Ctrl tumor cells.

**FGL2 inhibits p65 and STAT1/5 signals for CD103 induction**. GM-CSF regulates DCs development and differentiation by signal transduction through the PI3K/AKT, JAK2/STAT1/3/5, MAPK, and NF-κB pathways[30]. To investigate the specific pathway through which FGL2 suppresses GM-CSF-induced CD103 expression, we examined the changes in these signal pathways in CM-cultured bone marrow cells. The results show that FGL2KO-CM did increase TRAF6-NF-κB signaling, JAK2/STAT1/5 signaling, and p38 activation (Fig. 6a, Supplementary Figure 7). Blocking FGL2 using an FGL2-neutralized monoclonal antibody

in Ctrl-CM also increased activation of these three signaling pathways. Neutralizing GM-CSF in *FGL2KO*-CM reversed the activation of these signaling pathways (Fig. 6a). Thus, FGL2 suppresses GM-CSF-induced TRAF6-NF-κB signaling, JAK2/STAT1/5 signaling, and p38 activation in bone marrow cells.

Next, we asked whether these signaling pathways regulate CD103 expression on DCs. To address this question, we treated bone marrow cells with NF-κB activation inhibitor 6-amino-4-(4-phenoxyphenylethylamino) quinazoline (QNZ; EVP4593), IκB kinase activation inhibitor BAY11-7082, JAK2 activation inhibitor JSI-104, or p38 inhibitor PH797804. All of these inhibitors suppressed CD103 expression on DCs induced by *FGL2KO*-CM in a dose-dependent manner (Fig. 6b), showing that NF-κB signaling, JAK2 signaling, and p38 activation can indeed increase induction of CD103 expression in bone marrow cell culture. Therefore, these data indicate the underlying molecular mechanism by which FGL2 suppresses GM-CSF-induced CD103 expression on DCs or DC progenitors is mediated via inhibition of TRAF6-NF-κB, JAK2/STAT1/5, and p38 signal transduction (Fig. 6c).

**Concurrent GM-CSF and FGL2 in GBM predicts outcome**. The results presented in Figs. 5 and 6 collectively imply that the expression of GM-CSF in the absence of FGL2 may trigger an effective CD8⁺ T cell response in the brain against GBM. To ascertain whether this observation holds in GBM patients, we

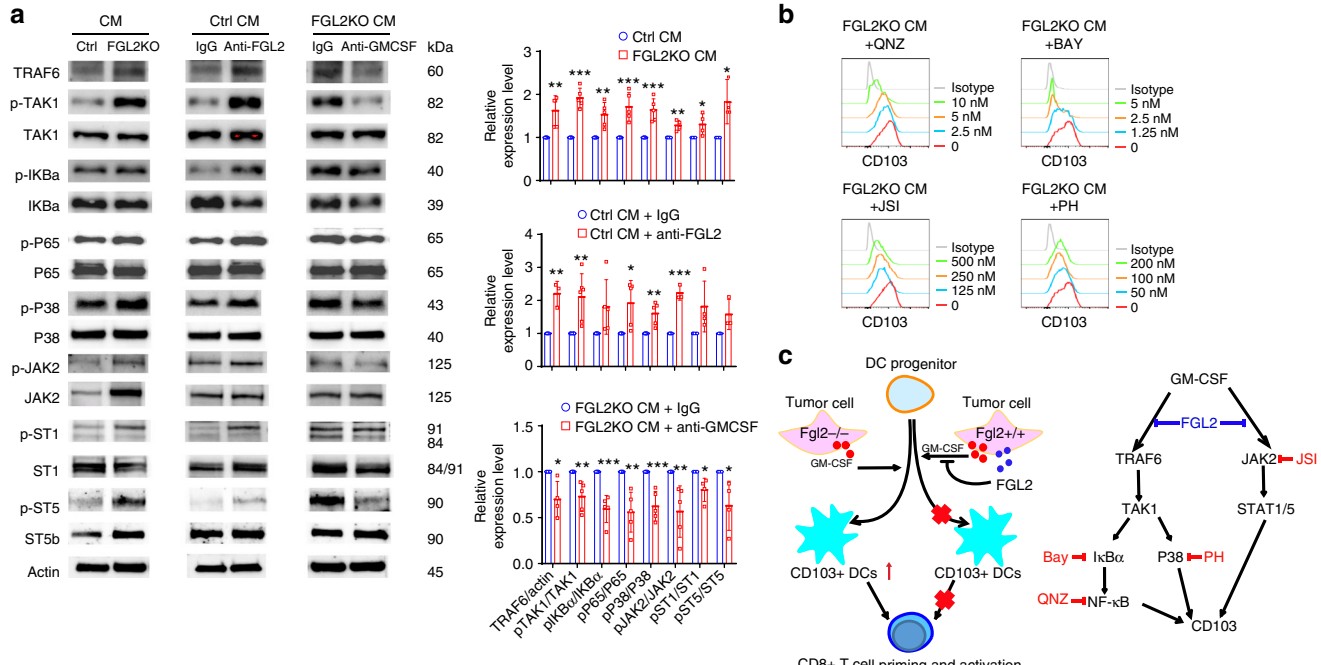

**Fig. 6** FGL2 suppresses NF-κBp65 and STAT1/5 signaling for CD103 induction. **a**, Expression levels of key proteins in the TAK1-NF-κB/p38 cascades and JAK2/STAT1/5 signaling were examined by western blotting in bone marrow cells cultured with conditioned medium (CM) from control tumor cells (*Ctrl*), *FGL2KO* tumor cells, *Ctrl*-CM + IgG, *Ctrl*-CM + anti-FGL2, *FGL2KO*-CM + IgG, or *FGL2KO*-CM + anti-GM-CSF for 3 days. Data were summarized as ratio changes from at least three independent experiments. The *t*-test was used to calculate the two-sided *P* values. Significant results were presented as \**P* < 0.05, \*\**P* < 0.01, \*\*\**P* < 0.001. **b** Representative FACS analysis of CD103 expression on CD11c⁺B220⁻ dendritic cells (DCs), which were pretreated for 1 h with IκBα inhibitor Bay 11-7085 (Bay), NF-κB inhibitor 6-amino-4-(4-phenoxyphenylethylamino) quinazoline (QNZ), JAK2 inhibitor JSI-124 (JSI), or p38 inhibitor PH797804 (PH) in bone marrow cells cultured with CM for 5 days. Plots are representative of three independent experiments. **c** Schematic illustration of cellular and molecular events underlying FGL2-regulated GBM progression. Tumor cells secreted GM-CSF and FGL2 simultaneously. GM-CSF-induced CD103⁺ DCs development was blocked in the presence of FGL2, so that CD8⁺ T cells were not primed and activated because of small CD103⁺ DC populations. Less CD103⁺ DC differentiation and subsequent lack of CD8⁺ T cell priming and activation resulted in GBM progression. For molecular signaling, TRAF6/TAK1/NF-κB/p38 signal and JAK2/STAT1/5 were activated in response to GM-CSF, contributing to CD103 induction. These signaling cascades were blocked by FGL2, thereby suppressing CD103 induction on DCs

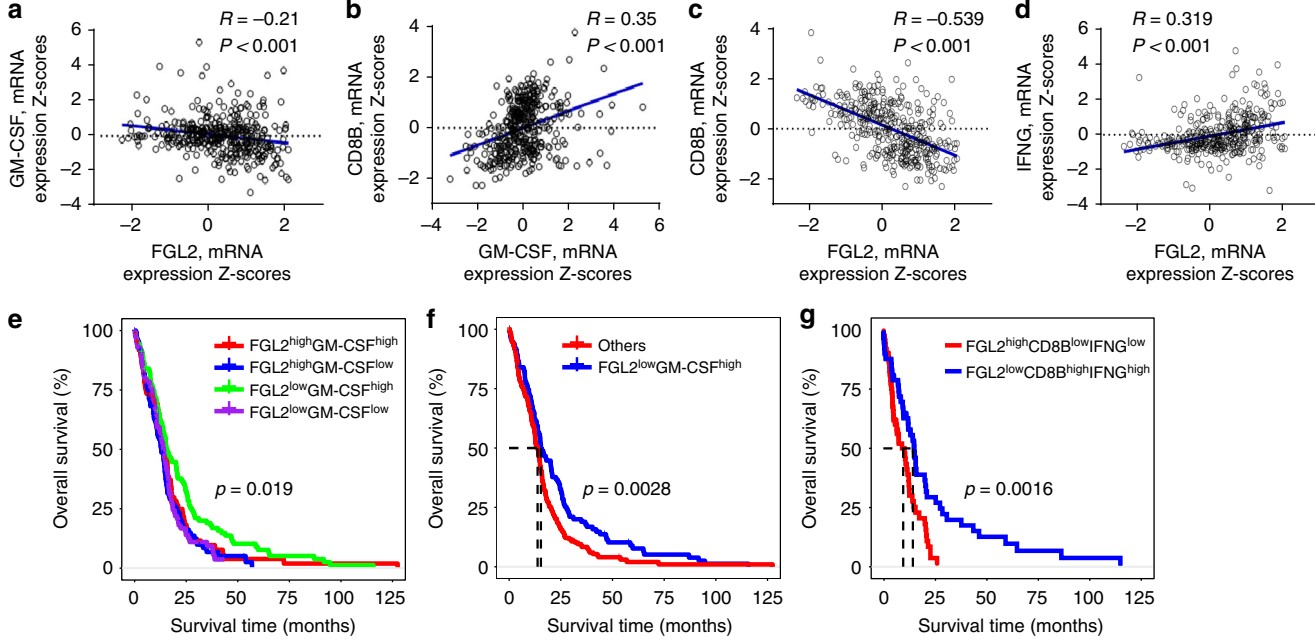

**Fig. 7** Expression level of *FGL2* and *GM-CSF* predicts outcome in GBM patients. Data on survival of GBM patients and gene mRNA expression data ($n = 401$ patients) were downloaded and retrieved from the TCGA data portal ([http://www.cbioportal.org/public-portal/]) Accessed between January 1, 2018, and March 15, 2018). The expression levels for both genes were categorized as high or low using median values as the cutoffs. **a–d** Correlations analysis of mRNA expression levels among *FGL2*, *GM-CSF*, *CD8B*, and *IFNG* in GBM tumors analyzed using the Pearson correlation coefficient. R = Pearson correlation coefficient. **e** Kaplan–Meier survival analysis for OS of GBM patients in the TCGA dataset grouped by expression levels of *FGL2* and *GM-CSF* mRNA. Patients' tumors were classified as *FGL2highGM-CSFlow*, *FGL2highGM-CSFhigh*, *FGL2lowGM-CSFlow*, or *FGL2lowGM-CSFhigh*. The log-rank test was used to compare overall survival among groups. **f** Kaplan–Meier survival analysis for OS of GBM patients in the *FGL2lowGM-CSFhigh* group vs. all others. The log-rank test was used to compare overall survival between groups. **g** Kaplan–Meier survival analysis for OS of GBM patients in the *FGL2lowCD8BhighIFNGhigh* group vs. those in the *FGL2highCD8BlowIFNGlow* group. The log-rank test was used to compare overall survival between groups. The statistical analyses were conducted using the R software package

analyzed the expression of these two genes and CD8$^+$ T cell-associated genes in the GBM dataset of The Cancer Genome Atlas (TCGA; dataset: Glioblastoma Multiforme, Provisional) to assess prognostic associations. As expected, increased expression levels of *GM-CSF* (hazard ratio [HR]: 0.854, $P = 0.008$) or *CD8B* (HR: 0.906, $P = 0.078$) in GBM was associated with longer overall survival (OS) (Supplementary Table 1), while increased *FGL2* expression was associated with poorer OS, though the difference was not statistically significant (HR: 1.113, $P = 0.071$) (Supplementary Table 1). There is a statistically significant positive correlation between *GM-CSF* and *CD8B* expression (Fig. 7b), and a strong negative correlation between *FGL2* and *CD8B* expression (Fig. 7c), suggesting that a low level of *FGL2* and a high level of *GM-CSF* may be associated with high CD8$^+$ T cell infiltration in tumors. When GBM samples were clustered by *GM-CSF* and *FGL2* expression levels (*FGL2highGM-CSFlow*, *FGL2highGM-CSFhigh*, *FGL2lowGM-CSFlow*, and *FGL2lowGM-CSFhigh*), OS differed significantly among the four clusters ($P = 0.019$) (Fig. 7e). Notably, patients with a concurrent decreased *FGL2* level and increased *GM-CSF* level had a longer median OS than other patients (median survival duration 15.4 vs 13.8 months, respectively, $P = 0.0028$) (Fig. 7f). Furthermore, patients with a concurrent decreased *FGL2* level and increased effector CD8$^+$ T cell signature (*CD8Bhigh IFNGhigh*) had a longer median OS than those with a concurrent increased *FGL2* level and decreased effector CD8$^+$ T cell signature (*CD8Blow IFNGlow*) (median survival duration 15.08 vs 10.45 months, respectively, $P = 0.0016$) (Fig. 7g). Overall, these results validated our mouse study results showing that concurrent high *GM-CSF* and low *FGL2* levels in tumors carry a more positive prognosis for GBM patients.

## Discussion

GBM is a deadly disease because of its aggressive growth, invasion of surrounding tissues, and poor immunogenicity. Here we show that knockout of FGL2 in tumor cells induces adaptive immune responses, resulting in 100% tumor suppression in GL261, DBT, and LLC brain tumor model systems. This adaptive antitumor immune response is characterized by increased differentiation of innate tumor antigen-presenting CD103$^+$/CD8a$^+$ DCs, which subsequently primes and induces tumor-specific CD8$^+$ cytotoxic T cells. This induced adaptive CD8$^+$ T cell response was also present in the periphery in the form of tumor-specific memory cells, although the *FGL2KO* tumor cells were implanted directly in the brain. These unexpected results suggest that FGL2 in tumor cells could serve as a potential therapeutic target for treating brain tumors and preventing tumor relapse. Supporting this conclusion, our previously published data showed that blocking FGL2 prolonged survival and reduced immune checkpoint CD39 and PD-1 expression and infiltration of CD39$^+$ Tregs and M2-type macrophages in the brain[8]. The current study puts forth the novel premise that tumor-elaborated FGL2 has a critical role in regulating CNS APCs.

APCs express major histocompatibility class II (MHCII) molecules and include DCs, macrophages, monocytes, B lymphocytes, and microglia in the brain[31]. Microglia and macrophages are the first innate immune cells that are recruited to brain tumors, but they express low levels of MHCII molecules[32]. Moreover, monocyte-derived macrophages in GBM assume an undifferentiated M0 state[33]. Monocytes exposed to malignant glioma cells adopt an MDSC-like immature phenotype[34]. Therefore, both GBM-associated microglia/macrophages

(tumor-associated macrophages [TAMs]) and MDSCs are "handicapped" APCs, which are efficient in antigen uptake but very weak in antigen presenting and cross-priming[32]. These "handicapped" APCs produce low levels of proinflammatory cytokines and lack expression of T cell co-stimulation molecules such as CD40, CD80, and CD86, and these APCs may result in T cell anergy instead of activation[32,35]. Therefore, it is not surprising that the number of TAMs/MDSCs positively correlates with human brain tumor progression[36,37].

It has been challenging to define the optimal DC population that may have an impact on clinical outcome. Analyses of tumor-bearing brains by us and others have shown that the infiltrating CD45$^+$CD11b$^+$ population includes CD11b$^+$ conventional DCs (cDCs), monocytes, monocyte-derived DCs induced by inflammation (moDCs), and macrophages; CD45$^+$CD11c$^+$CD11b$^-$ cells include plasmacytoid DCs (pDCs) and CD11b$^-$ cDCs; and CD45$^+$ CD11b$^-$CD11c$^-$MHCII$^+$ cells comprise B cells primarily. Among the three major categories of DCs in brain tumors, pDCs are poor APCs because of low levels of MHCII expression, but they are beneficial in enhancing cross-priming when activated[38,39]; moDCs exhibit less T cell priming functions than cDCs, yet they accounted for 30% of the total DC population and thus may influence the adaptive antitumor immune response[40,41]; and cDCs are highly effective professional APCs located in the meninges and peritumor area[41]. However, the specific subpopulation of cDCs that induce tumor-specific CD8$^+$ T cells has not been defined until now. DCs exist as phenotypically and functionally distinct subpopulations within tissues, distinguished by characteristic patterns of surface marker expression[29,42]. Mounting evidence from peripheral tumors shows that Batf3-dependent cDCs are the most critical DCs in cross-presentation of tumor cell-associated antigens and priming tumor-specific T cell immunity in both mice and humans[5,18–20,22,43]. Indeed, tumor-isolated CD103$^+$ DCs, which were induced in the presence of Batf3, have a markedly higher efficiency in cross-presenting tumor antigens than other DC subsets, macrophages, monocytes, and neutrophils[19]. Intratumoral CD103$^+$ DCs traffic to TDLNs by expressing CCR7 to prime naive CD8$^+$ T cells, causing tumor-specific CD8$^+$ T cells to increase in number and infiltrate tumors[5]. Tumor-residing CD103$^+$ DCs are necessary for CD8$^+$ effector T cell recruitment by producing CXCL9/10 chemokines[18]. Moreover, CD103$^+$ DCs, after activation, produce high levels of Th1-differentiating cytokines, such as IL-12, to provide essential signals for generation of memory CD8$^+$ T cells[19,22]. Thus, CD103$^+$ DCs have a crucial role in the initial antigen presentation and priming, effector, and memory phases required to induce an antitumor adaptive T cell response.

CD103$^+$ cDCs have been characterized in peripheral tumors but not previously in brain tumors. Our findings in this study show the presence of both CD103$^+$ and CD8a$^+$ DC populations in brains and TDLNs of brain tumor-bearing mice, and total DC populations with higher proportions of CD103$^+$/CD8a$^+$ cells have activated OT-I T cells in vitro and induced OT-I T cell proliferation in vivo, showing that CD103$^+$/CD8a$^+$ DC populations in the brain have roles in antigen-presenting and cross-priming effects, as observed in peripheral tumors.

Batf3 is a key transcription factor in regulation of CD103$^+$/CD8a$^+$ DC differentiation[29]. Indeed, Batf3$^{-/-}$ mice showed impaired rejection of FGL2KO tumors because of significantly reduced CD103$^+$/CD8$^+$ DCs in the brain and TDLNs, showing impairment in initiating the CD8$^+$ T cell response against tumor cells in the absence of Batf3. Supporting our results are findings by Quitana et al.[44] that, although CD45$^+$CD11c$^+$CD11b$^{low}$Clec9A$^+$ DC populations (Clec9A is specifically expressed by CD8a$^+$/CD103$^+$ DCs in mice) were

increased in noninjured brains when treated with FLT3L, this effect was abolished in the brains of Batf3$^{-/-}$ mice. Therefore, the Batf3-dependent CD103$^+$ DC population in the brain is critical for CD8$^+$ T cell-dependent elimination of brain tumors.

Both GM-CSF and Flt3L are essential growth factors for DC development in vitro and in vivo. Culture of bone marrow cells with FLT3L in vitro yields cDCs and pDCs[45]. FLT3L treatment in vivo promoted the differentiation of CD103$^+$ DC progenitors in the bone marrow and their expansion at the tumor site by enforcing IRF8 signaling, resulting in accumulation of immature CD103$^+$ DCs and accumulation of Treg cells at the tumor site. Thus, FLT3L on its own has limited ability to improve antitumor immune responses in vivo[22]. GM-CSF promotes the differentiation of myeloid cells and has an important role in differentiation, proliferation, and activation of DCs[30,46]. Culture of bone marrow cells with GM-CSF yields macrophages and DCs[47]. GM-CSF is essential for tumor cDC development. GM-CSF receptor-depleted bone marrow cells failed to reconstitute cDCs in tumors when transferred to tumor-bearing mice[19]. Knockout of GM-CSF in mice yields strong reduction of tissue-resident CD103$^+$ DCs and migratory CD103$^+$ DCs[19]. Likewise, absence of the GM-CSF receptor on lung DCs abrogated the induction of CD8$^+$ T cell immunity after immunization with particulate antigens[48].

The direct linkage of GM-CSF expression and CD103$^+$ DC generation suggests that a GM-CSF vaccine should be effective in achieving an antitumor response. However, after over two decades of effort, clinical trials of different types of GM-CSF cancer vaccines have had mixed results[49–51], and most human clinical trials have been disappointing[52–54]. Our study suggests that one possible reason for this is the presence of FGL2 in tumor cells, which inhibits CD103$^+$ DC differentiation and has not been previously accounted for. Our results clearly show that FGL2 expressed by tumor cells reduces CD103$^+$ DC differentiation. Therefore, future GM-CSF-based cancer vaccines may require the use of FGL2 depletion or blockade or patient stratification based on the absence of tumor-expressed FGL2. Of course, this view requires further validation because we have tested only GBM cells and a lung cancer cell line thus far. However, knockout or depletion of FGL2 may not only increase GM-CSF-mediated CD103$^+$ DC differentiation but probably also reduces MDSC and TAM levels at the same time. Indeed, our previous data show that overexpression of FGL2 increased MDSC and M2 cell levels in GBM[8]. This increase in MDSCs and M2s may be another factor that negatively affects the development of a GM-CSF vaccine. Supporting this view, others have found that the GM-CSF-mediated bone marrow response is biphasic, with a low level reducing MDSC and macrophage numbers and activity and a high level having the opposite effect[47,55].

It is well known that regulation of DC differentiation by GM-CSF depends on the activation and signaling of PI3K/PKB, JAK/STAT, MAPK, and canonical NF-κB. GM-CSF induces proliferation and prevents cell death, mainly by activating PKC/PKB and MAPK signaling, and induces differentiation and inflammatory responses by MAPK, NF-κB, and STAT5 signaling[56]. However, the signaling pathways underlying the induction of CD103$^+$ cDC differentiation by GM-CSF are unclear. STAT5, Batf3, and IRF8 are known to facilitate GM-CSF-induced CD103$^+$ DC development[28,57], but no other signaling pathways are clearly linked yet. Here we show that GM-CSF induces TAK1/NF-κB, p38, and JAK2/STAT5/STAT1 signal transduction in order to induce CD103 expression. Adding to these novel discoveries is the crucial finding that the presence of subtle FGL2 levels from tumor cells can significantly reduce the intensity of these CD103$^+$ DC-inducing signals. However, PI3K/AKT and ERK signaling did not participate in suppression of GM-CSF signaling by FGL2. Thus, TAK1/NF-κB signaling, JAK2/STAT1/

5 signaling, and p38 signaling are three pathways regulated by FGL2 to inhibit GM-CSF-mediated CD103 expression. This novel discovery could help shine new light on the biology of CD103[+] DC immunity.

## Methods

**Human samples.** Sample collection was conducted under protocol #LAB03-0687, which was approved by the Institutional Review Board of The University of Texas MD Anderson Cancer Center, after written informed consent was obtained. Patients' tumors were graded by a neuropathologist according to the World Health Organization classification[58]. Resected tumor tissues from newly diagnosed primary GBM patients were used in this study.

**Animals.** C57BL/6J, BALB/c, CD45.1 (B6.SJL-Ptprc[a] Pepc[b]/BoyJ, 002014), OT-I (C57BL/6-Tg(TcraTcrb)1100Mjb/J, 003831), NSG (NOD.Cg-Prkdc[scid] Il2rg[tm1Wjl]/SzJ, 005557), CD11c-DTR (B6.FVB-1700016L21Rik[Tg(Itgax-DTR/EGFP)57Lan], 004509), and Batf3[−/−] (B6.129 S(C)-Batf3[tm1Kmm]/J, 013755) mice were obtained from the Jackson Laboratory (Bar Harbor, ME, USA). FGL2[−/−] mice were a gift from Dr. Gary Levy[59] (Toronto General Hospital/Research Institute, Toronto, ON, Canada). The RCAS/Ntv-a system was used to generate CD8[−/−] mice[60]. In brief, the Ntv-a transgene (avian cell surface receptor for subgroup A avian leukosis virus under the control of a glial progenitor-specific promoter derived from the human NES gene) and the CD8α-targeted allele were moved from their respective genetic backgrounds onto the C57BL/6 background by marker-assisted backcrossing to yield Ntv-a/CD8α[−/−] and Ntv-a/CD8α[+/+] mice. All mice were aged 6 to 8 weeks when the experimental procedures began. All mice were maintained and treated in accordance with guidelines approved by the Institutional Animal Care and Use Committee (IACUC) at MD Anderson.

**Cells.** Glioblastoma (GBM) stem cell (GSC) lines (GSC7-2, GSC11, GSC20, GSC28, pGSC2, reGSC1, GSC7-11, and GSC8-11) were obtained through Dr. Frederick F. Lang[60,61], and HMVEC-L cells were provided by Dr. Schadler Keri (both of MD Anderson). The mouse glioma GL261 cells were obtained from the National Cancer Institute (Rockville, MD, USA), and cortical HCN-1A neurons from ATCC (Manassas, VA, USA). The DBT cells were kindly provided by Dr. Leonid Metelitsa (Baylor College of Medicine, Houston, TX, USA). All cells were treated with mycoplasma removal agent (BUF035, BIO-RAD, Hercules, CA, USA) to remove mycoplasma from cell cultures before experiments.

**FGL2 knockout.** Mouse FGL2 guide RNA (gRNA) sequences GCTGCCGCACTGCAAGGATG and GTGCTCCTCCACCGCTCGGC were cloned into the **pX458** vector for CRISPR-associated protein 9 and gRNA expression (provided by Dr. Feng Zhang and colleagues at the Massachusetts Institute of Technology, Cambridge, MA, USA). Mouse glioma GL261 cells, mouse astrocytoma-derived DBT cells, and lung cancer LLC cells were transfected with both guide plasmids by electroporation. The cells were grown for 2 weeks, and green fluorescent protein (GFP)-positive cells were isolated from cultures of each cell type and seeded at 1 cell per well to yield single-cell clones. The knockdown/knockout of FGL2 expression in clonal cells was confirmed by western blotting and sequencing of PCR fragment cloning of stable clones. The scramble control gRNA sequence was CGCTTCCGCGGCCCGTTCAA.

**Mouse model systems.** For the orthotopic GBM mouse models, GL261 cells, both controls (GL261-Ctrl) and FGL2 knockout (GL261-FGL2KO), DBT cells (DBT-Ctrl and DBT-FGL2 knockdown [KD]), and LLC cells (LLC-Ctrl and LLC-FGL2KO) were treated with mycoplasma removal agent and collected in logarithmic growth phase, washed twice with phosphate-buffered saline solution (PBS), and mixed with an equal volume of 3% methylcellulose in PBS. Cells ($5 \times 10^4$, $2.5 \times 10^5$, or $1 \times 10^6$ in a total volume of 5 μL) were injected intracerebrally into the mice[61]. The investigator was blinded to cell lines for tumor implantation. The mice were observed daily. When the mice showed signs of neurological compromise, they were humanely killed with $CO_2$.

For the subcutaneous tumor model, tumor cells were suspended in PBS, and 30 μL ($1 \times 10^5$) were injected subcutaneously into one flank of each mouse. Tumors were measured twice per week. Mice that showed signs of morbidity, high tumor burden, or skin necrosis were immediately compassionately killed according to IACUC guidelines. Tumor volumes were calculated using the formula $\pi(\text{length} \times \text{width}^2)/8$. The length represents the longest axis, and the width is at right angles to the length. All animal experiments were repeated at least twice.

**In vivo bioluminescence imaging.** On day 1 and day 7 after DBT tumor-cell implantation, mice were injected intraperitoneally with 150 mg/kg of D-luciferin in PBS 10 min prior to imaging. Imaging was performed with a charge-coupled device camera (IVIS 100; Xenogen-Caliper, Alameda, CA, USA). Total photon flux (photons/s) was measured from a fixed region-of-interest over the skull using Living Image and IgorPro software (Wavemetrics, Portland, OR, USA)[62].

**CD4, CD8, and NK cell depletion.** Anti-CD8α (clone 2.43), anti-NK1.1 (clone PK136), anti-CD4 (clone GK1.5) antibody, or their matched IgG isotype was injected at a dose of 200 μg per mouse, 2 days before tumor inoculation and every 4 days thereafter. These antibodies were purchased from BioXCell (West Lebanon, NH, USA).

**FTY720.** Wild-type mice were injected intraperitoneally with 1 mg/kg fingolimod (FTY720) (Cayman Chemical, Ann Arbor, MI, USA) in sterile saline solution 1 h before implantation with GL261-FGL2KO tumor cells. The mice were maintained on drinking water containing 2 μg/mL FTY720 for the duration of the experiments[22].

**OT-I proliferation in vivo.** CD45.1 mice were implanted with control GL261 ovalbumin-expressing cells (GL261-Ctrl-OVA) or GL261-FGL2KO-OVA tumor cells ($2.5 \times 10^5$ per mouse). CFSE-labeled ovalbumin-specific T cell receptor transgenic (OT-I) cells ($5 \times 10^6$) were injected intravenously 2 days later. Brain tissue and TDLNs were collected on day 7 and subjected to FACS assay for the CD45.2[+]CD8[+]CFSE[+] cell population[22].

**Bone marrow dendritic cell culture.** Femurs and tibiae from C57BL/6 mice were harvested under sterile conditions and bone marrow was collected by flushing. The red blood cells were removed by lysis. Bone marrow cells ($2 \times 10^6$/mL) were cultured in RPMI1640 medium supplemented with 10% heat-inactivated fetal bovine serum (FBS; Biochrom, Holliston, MA, USA), penicillin/streptomycin, 1 mM sodium pyruvate, and 50 μM β-mercaptoethanol. Conditioned medium (CM) produced by FGL2KO or control tumor cells was added at a volume ratio of 1:1. On day 5, equal volumes of pre-warmed fresh medium and CM were added, and cultures continued for another 4 days. Non-adherent cells in the supernatants were collected and re-plated with fresh medium supplemented with 50 ng/mL hFLT3L (Peprotech, Rocky Hill, NJ, USA) and CM for 6 more days. Bone marrow DCs were collected at day 5 or 15 for analyzing the expression of CD103, Clec9A, and MHCII on CD11c[+]B220[−] cells. In some experiments, bone marrow cells were pretreated with IκBα inhibitor Bay 11-7085 (5, 2.5, or 1.25 μM; Invitrogen, Carlsbad, CA, USA), NF-κB activation inhibitor 6-amino-4-(4-phenox-yphenylethylamino) quinazoline (QNZ; 10, 5, or 2.5 nM; Cayman Chemical), STAT3-JAK inhibitor JSI-124 (500, 250, or 125 nM; Cayman Chemical), or p38 MAPK inhibitor PH797804 (200, 100, or 50 nM; Selleck Chemical, Houston, TX, USA) for 1 h in culture with RPMI1640 medium[63–65]. For GM-CSF culture, bone marrow cells were cultured in RPMI1640 medium supplemented with recombinant murine GM-CSF (rmGM-CSF, 10 ng/mL; 315-03; Peprotech) for 7 days (fresh medium with GM-CSF added on days 3 and 5)[28,47]. In some experiments, bone marrow cells were pretreated with IgG (5 μg/mL), anti-FGL2 (5 μg/mL; prepared in-house by the Monoclonal Antibody Core Facility, MD Anderson Cancer Center), anti-GM-CSF (5 μg/mL; MP122E9, MAB415; R&D Systems, Minneapolis, MN, USA), or recombinant FGL2 (rFGL2; 200 ng/mL; R&D Systems) for 1 h. Three independent experiments were repeated for in vitro experiments.

**OT-I activation assay.** OT-I cells were isolated from the spleen and lymph nodes of OT-I mice and purified with a MojoSort Mouse CD8 T cell Isolation Kit (480008; BioLegend, San Diego, CA, USA). DCs were isolated from FGL2KO-OVA or Ctrl-OVA tumors from tumor-bearing mice with an EasySep Mouse Pan-DC Enrichment Kit Immunomagnetic negative selection cell isolation kit (19763; StemCell Technologies, Vancouver, BC, Canada). Isolated DCs ($5 \times 10^4$) were co-cultured with naive CFSE-labeled OT-I cells (5 μM CFSE for 5 min) at a 2:1 ratio in 96-well round-bottom tissue culture plates[19,47]. After 24 h of co-culture, the OT-I cells were collected and analyzed for CD69 expression by flow cytometry.

For in vitro cross-presentation assay, CM-cultured DCs were collected and resuspended at $2.5 \times 10^5$/mL with 100 μg/mL soluble OVA protein to pre-incubate for 2 h. After two washes, $5 \times 10^4$ CFSE-labeled OT-I cells were added to the co-culture for 24 h. The OT-I cells were collected and analyzed for CD69 expression by flow cytometry[66].

For T cell receptor activation, 96-well plates were coated with 10 μg/mL CD3 (100 μL/well; clone 145-2C11/37.51; BioLegend) and incubated for 4 h at 37 °C. CFSE-labeled OT-I cells ($2 \times 10^5$ in 100 μL culture medium) were added to each well. Conditioned medium (100 μL) was added to some wells as treatment. After 72 h, the contents of the well were subjected to FACS assay to detect CD25 expression and CFSE dilution in OT-I cells[67].

**In vitro cytotoxic T lymphocyte assay.** Target GL261 tumor cells were labeled with 1.5 μM CFSE (2× concentration) for 10 min. Effector CD8[+] T cells were isolated from tumor-bearing mice. Target cells and effector CD8[+] T cells were co-cultured at ratios of 1:5, 1:20, and 1:80 for 6 h. CFSE[+] propidium iodide (PI)-positive tumor cells were detected by flow cytometry.

For the OT-I killing assay, naive OT-I cells were pre-activated with 2 μg/mL ovalbumin peptide SIINFEKL (OVA257-264) for 24 h. Tumor cells ($10^6$ cells/mL) were incubated with calcein AM in a final concentration of 2 μM for 30 min. Target cells and effector CD8[+] T cells were co-cultured at ratios of 1:0 1:1, 1:5, 1:20, and 1:80 for 6 h. The supernatants were collected and fluorescence was quantified at excitation filter 485 nm and emission filter 530 nm.

**CD8$^+$ T-cell survival assay**. Naive CD8$^+$ T cells ($1 \times 10^6$/mL) were activated with plate-bound CD3 and CD28 antibodies for 2 days. Recombinant murine IL-2 (500 U/mL) was then added and the cells were cultured for 3 more days. Cells were collected, washed, and plated in CM from GL261-*Ctrl* or GL261-*FGL2KO* tumor cells and devoid of IL-2 and cultured for 5 more days. The cells were then stained with PI to isolate surviving cells[68,69].

**Migration assay**. OT-I CD8$^+$ T cells in suspension were stained with 2.5 μM Cell Tracker Violet BMQC Dye (C10094, Invitrogen) for 30 min. CM from tumor cells (0.75 mL of medium containing 5% FBS) were plated in the lower wells of a chamber assay plate, and OT-I cells ($2.5 \times 10^5$ cells in 0.1 mL of medium containing 1% FBS) were placed in the upper wells of the insert. After co-incubation for 1, 2, 4, or 6 h, cells were collected from the bottom wells and subjected to flow cytometry to identify BMQC$^+$ cells[70].

**Flow cytometry**. Patient tumor tissues were minced and enzymatically digested to obtain single-cell suspensions for staining. Myelin debris was removed by using Myelin Isolation Beads (130-104-262; Miltenyi Biotec, Bergisch Gladbach, Germany). Mouse brain tissues and tumor-draining lymph nodes (TDLNs) were minced and enzymatically digested to obtain single-cell suspensions. Brain-infiltrating leukocytes were isolated according to a previously published protocol[71]. Briefly, each single-cell suspension was subjected to centrifugation through a 30% Percoll gradient at 7800×*g* for 30 min. The leukocyte layer was collected and subjected to centrifugation on a discontinuous Ficoll-Paque Plus gradient to select and purify leukocytes. Fc receptors were blocked using anti-CD16/CD32 (2.4G2, BD Biosciences, San Jose, CA, USA). The antibodies were purchased from Biolegend, eBioscience (Thermo Fisher Scientific, Waltham, MA, USA), BD Biosciences, or R&D Systems: αCD45 (30-F11), αCD45.2 (104), αCD3 (17A2), αCD4 (GK1.5), αCD8α (53-6.7), αCD11b (M1/70), αCD11c (N418), αCD45R/B220 (RA3-6B2), αCD103 (2E7), αClec9A (42D2), αI-A/I-E (M5), αLy-6G/Gr-1 (RB6-8C5), αNK1.1 (PK136), αTer119 (TER119), αCD25 (3C7), αCD69 (H1.2F3), αFoxp3 (FJK-16s), αIFN-γ (XMG1.2), αGZMB (NGZB), αCD31 (9G11), αCD45 (2D1), and αGFAP (GA5). The cell surfaces were stained by a standard protocol. For intracellular staining, cells were fixed, permeabilized, and incubated with IFNγ and granzyme B. Staining for nuclear transcription factor Foxp3, T-bet, and Eomes was performed according to a protocol provided by eBioscience. Stained cells were isolated by flow cytometry and the results analyzed by FlowJo software.

**Western blotting**. To quantify expression of specific proteins, cells were subjected to lysis in RIPA buffer (50 mM Tris-HCl, pH 7.4; 1% NP-40, 0.25% sodium deoxycholate, 150 mM NaCl, 1 mM EDTA) supplemented with 50 mM NaF, 20 mM β-glycerophosphate, and a complete protease inhibitor cocktail (Roche Diagnostics, Indianapolis, IN, USA). Protein concentrations were determined using the bicinchoninic acid reagent (Thermo Fisher Scientific). Proteins were separated by sodium dodecyl sulfate 10% polyacrylamide gel electropheresis and transferred to polyvinylidene fluoride membranes. Membranes were incubated overnight at 4 °C with primary antibodies. After washing, the blots were incubated with the appropriate horseradish peroxidase-conjugated secondary antibody and processed to detect electrochemiluminescence signals. The anti-FGL2 antibody was then applied for immunoblotting (monoclonal anti-FGL2 antibody produced in mouse, 1:1000). The antibodies were purchased from Cell Signaling Technology (Danvers, MA) or Santa Cruz Biotechnology (Dallas, TX): αTRAF6 (D21G3), αp-TAK1 (Ser412), αTAK1 (D94D7), αp-IKBa (14D4), αIKBa (L35A5), αp-P65 (93H1), αP65 (D14E12), αp-P38 (D3F9), αP38 (D13E1), αp-JAK2 (D15E2), αJAK2 (D2E12), αp-ST1 (D4A7), αST1 (B-9), αp-ST5 (C11C5), αST5b (G-2), β-Actin (13E5), and αGAPDH (D16H11). ImageJ software was used to quantify blots. GAPDH was used as a control, and results were quantified by calculating the band intensity of protein X relative to GAPDH. Quantities are summarized as ratio changes from at least three independent experiments.

**Immunohistochemistry and immunofluorescence**. Mouse brains were formalin-fixed and paraffin-embedded, and 4-μm sections were cut for hematoxylin and eosin (H&E) staining. For immunofluorescence, sequential staining was used. First, GBM tissue sections were stained with mouse anti-FGL2 antibody (1:100, H00010875; Novus Biologicals, Littleton, CO, USA) overnight at 4 °C, followed by 1 h incubation with anti-mouse Alexa Fluor 647-conjugated antibody. The sections were blocked with Fab fragment anti-mouse IgG (15-007-003; Jackson ImmunoResearch Laboratories, Inc., West Grove, PA, USA) for 1 h, and then stained with rabbit anti-CD31 antibody (bs-0468R, 1:100; Bioss Inc., Woburn, MA, USA), mouse anti-CD45 antibody (MAB1430, 10 μg/mL; R&D Systems), or Alexa Fluor 488-conjugated GFAP (53-9892, 5 μg/mL; Thermo Fisher Scientific) overnight at 4 °C, followed by incubation with Alexa Fluor 488-conjugated anti-mouse/rabbit antibody (1:300; Invitrogen) for 1 h. ProLong Gold Antifade Mountant with 4′-diamidino-2-phenylindole (DAPI; Thermo Fisher Scientific) was used as the mounting medium. Slides were further processed for imaging analysis using an Olympus Fluoview FV1000 microscope.

**GM-CSF and FLT3L assays**. Medium conditioned by *FGL2KO* (*FGL2KO*-CM) or control (*Ctrl*-CM) cells was collected after 48 h culture at a cell density of $2 \times 10^6$.

The expression levels of GM-CSF and FLT3L in the CM were determined by using ELISA kits, the Mouse GM-CSF Quantikine ELISA Kit (MGM00; R&D Systems) and the Mouse Flt-3 ligand PicoKine ELISA Kit (EK0355; Boster Biological Technology, Pleasanton, CA, USA), according to the manufacturer's instructions.

**The Cancer Genome Atlas data analysis**. The Cancer Genome Atlas (TCGA) mRNA expression data were accessed via the cBioPortal for the Cancer Genomics Web site [http://www.cbioportal.org/public-portal/]. The provisional mRNA expression z-scores (microarray) for *FGL2*, *GM-CSF*, *CD8B*, and *IFNG* genes with matched outcome data (OS) were used. The z-score (positive or negative) represents a normalized expression level relative to non-neoplastic tissue controls[72,73]. By recognized practice, the median gene expression z-score of the complete cohort (401 patients) was utilized as a cutoff value for classification of patients into high and low-expression groups for each individual gene of interest, including *FGL2* (cutoff value 0.33), *GM-CSF* (cutoff value −0.02), *CD8B* (cutoff value 0.01), and *IFNG* (cutoff value −0.24). According to the cutoff value of each gene, the patients with expression z-score values higher than the median were classified into the "high-expression" group (denoted as *FGL2*$^{high}$, *GM-CSF*$^{high}$, *CD8B*$^{high}$, and *IFN-G*$^{high}$), and the patients with expression z-scores less than the median were classified into the "low-expression" group (denoted as *FGL2*$^{low}$, *GM-CSF*$^{low}$, *CD8B*$^{low}$, and *IFNG*$^{low}$). We considered various groups of patients with different combined expression levels of multiple genes, for example, comparing patients in the group with combined *FGL2*$^{high}$, *CD8B*$^{low}$, and *IFNG*$^{low}$ to patients with *FGL2*$^{low}$, *CD8B*$^{high}$, and *IFNG*$^{high}$. A univariate Cox proportional hazards model was used to assess the association between gene expression levels (both individual genes and combined genes) and OS. The hazard ratio (HR) and 95% confidence interval (CI) were calculated and reported. Pearson correlation coefficients were calculated between *CD8B*, *GM-CSF*, *IFNG*, and *FGL2* gene expressions. The statistical analyses were conducted using R package software (R Development Core Team, Version 3.3.2). A *P* value < 0.05 was considered statistically significant. Data were accessed from the cBioPortal [http://www.cbioportal.org/public-portal/] between January 1, 2018, and March 15, 2018.

**Statistical analysis**. All quantified data are presented as mean ± standard deviation (s.d.) or as indicated. Differences in the experimental means for flow cytometric and cytokine values were considered significant if *P* was <0.05 as determined by one-way analysis of variance (ANOVA) or *t*-test. A two-way ANOVA was used to analyze tumor volume differences between groups. Normality of data was assessed using the Shapiro–Wilk test of normality. Assumption for homogeneity of variances was assessed using the Levene's test. For the data for which the underlying assumptions were not met, we removed outliers, used a test accounting for unequal variances (e.g., Welch's *t*-test, Welch's ANOVA), or used a nonparametric test such as the Wilcoxon rank-sum test. Tukey's multiple comparison test was used for pairwise comparisons in the ANOVA analysis. Differences in survival curves were analyzed by Kaplan–Meier analysis and the log-rank test, and differences were considered significant at *P* < 0.05. All data shown are representative of three independent experiments acquired in triplicate (in vitro) or at least two independent experiments (in vivo). Unless otherwise noted, statistical tests were two-sided and statistical analyses were conducted using GraphPad Prism 7 software (GraphPad Software, La Jolla, CA) and R package software (R Development Core Team, Version 3.3.2).

**Reporting summary**. Further information on experimental design is available in the Nature Research Reporting Summary linked to this article.

## Data availability

The authors declare that all data supporting the findings of this study are available within the paper and its Supplementary Information files or from the corresponding author upon reasonable request. The TCGA data referenced during the study are available in a public repository from the cBioPortal website [https://www.cbioportal.org/public-portal/].

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

## Acknowledgements

We thank Mrs. Kathryn L. Hale (Department of Scientific Publications) and Dr. David M. Wildrick, MD Anderson Cancer Center, for editorial assistance. We also thank Dr. Meng Zhou and Dr. Jianzhong Su for their suggestions for TCGA analysis and Dr Willem W Overwijk for valued suggestions on this project. GSCs were provided by Dr. Frederick Lang, HMVEC-L cells from Dr. Keri Schadler, and DBT cells from Dr. Leonid Metelitsa. FGL2$^{-/-}$ mice were provided by Dr. Gary Levy. This research was supported by grants from the U.S. National Institutes of Health (CA203493, CA127001) to Dr. Shulin Li and Dr. Frederick Lang. The Monoclonal Antibody core facility is supported by National Institutes of Health Cancer Center Support Grant P30CA016672.

## Author contributions

S.L. and A.B.H. initiated the original idea for this work. J.Y. and S.L. planned the experiments. J.Y., Q.Z., L.-Y.K., X.X., J.X., and L.H. performed the experiments. K.G. and M.O. collected clinical samples. J.Y. collected data and performed analyses. J.W. analyzed TCGA data. J.Y., A.B.H., and S.L. prepared the manuscript draft with input from all authors. S.S.W., S.C.S., R.E.D., and G.R. provided optimized protocol and experiment materials. All authors provided critical feedback and helped shape the research, analysis, and manuscript.

## Additional information

**Competing interests:** The authors declare no competing interests.

