## [Peer Review File · Nature Communications]

Reviewers' Comments:

Reviewer #1:

Remarks to the Author:

In this manuscript the authors demonstrate that expression of Fibrinogen-like protein 2 (FGL2) suppresses the immune response in glioblastoma using a combination of NSG mice, CD8-deficiency, and CD8-depletion. This finding is convincing, but unfortunately the proposed mechanism of action involving suppressed maturation of dendritic cells (DCs) is not well supported by the data. If the authors can address this point, or expand their findings to show that FGL2 directly suppresses T cell activation, then the manuscript should be of sufficient interest and quality for publication. Details are below:

Major Points:

- 1) In figure 4 there appear to be almost no DCs in the LNs and tumors. This is possibly due to gating out the CD11b+ subset, but also gating out many of the CD103+ DCs (which express moderate amounts of CD11b). Finally, there should be no CD8a DCs in peripheral tissues (unless TLNs are forming). Are these CD8a+ DCs the same population as the CD103+ DCs in this analysis? CD103+ DCs do not normally express CD8a. A full gating schema for LNs and tumors should be shown in the supplemental figures to demonstrate that the authors are able to accurately identify these populations.
- 2) Batf3-deficient mice are unable to mount an effective CD8 response, but this does not necessarily imply that the cells are being functionally regulated by the pathway of interest. For example, anti-PD-1 also fails in Batf3-deficient mice, even though DCs are not a target of this therapeutic approach. Indeed the authors show a direct effect of FGL2 on CD8+ T cells during CD3 stimulation in vitro. The OT-I adoptive transfer and FTY720 experiments imply a role for naïve T cell activation, but again do not discriminate between impacts on DCs or T cells.
- 3) There are no studies performed to indicate that the DCs treated with FGL2 are less functional, the slight reduction in CD103 expression notwithstanding. The data from Fig 5a suggests these differences may be due to artefactual gating out of activated cells (that are simply appearing B220 high) or altered maturation of plasmacytoid DCs.
- 4) Generation of CD103+ DCs from bone marrow has been reported to take about 15 days using a combination of GM-CSF and FLT-3L (See Mayer et al. Blood 2017). Little to no CD103 is found on cells matured with either cytokine alone even after 7-10 days. It is therefore surprising that the authors are showing data using conditioned medium that induces high CD103 expression in 5 days. Additional phenotyping is required to claim that these are CD103+ DCs.
- 5) The data in Figure 1 suggests that endothelial cells and leukocytes may express a significant fraction of the FGL2 in the tumor. Although the functional data in the subsequent figures indicates an important role for FGL2 expression by tumors cells, the related text should be clear that the microenvironment is also a significant source. It would also be better to report the percentage of the subset positive for FGL2 for the flow cytometry data (e.g. 70% of CD31+ cells, not 1.8%).

Minor Issues:

- 1) Fig S6D would benefit from a proliferation analysis.
- 2) In Figure 7B/C the impact on survival is not very convincing. Focusing on the impact of FGL2 expression on CD8/IFNG (Fig 7d) would be more informative. However, the survival differences without FGL2 need to be shown, otherwise it is unclear if this adds anything to segregating out patients with high CD8/IFNG expression.

3) The discussion covers a wide range of topics, many of which are only tangentially related to the paper (e.g. B cells, DC vaccination).

4) There is no data to indicate that FGL2 is an "immune checkpoint regulator". These words should be removed from the title and text. A suggested title might be "FGL2 promotes glioblastoma progression by suppressing CD103+ dendritic cell differentiation" would be appropriate.

Reviewer #2:

Remarks to the Author:

The manuscript by Yan et al. aims to provide evidence that FGL2 expressed by tumor cells reduces tumor growth by stimulating an immune response against tumors. They further suggest that this is achieved by FGL2 mediated differentiation of CD103+ dendritic cells and the stimulation of an anti-tumor T cell response.

Over the last years evidence has been presented that tumor cells encode a series of molecules that inhibit immune responses against the tumors themselves. These molecules do not exert their effects through modifying the activity of MDSCs or Tregs. A number of these molecules belong to the immune system, such as PD-L1, while others are unrelated to molecules known from the immune system. Tumors also express molecules that directly inhibit NK cells, such as LDH5 and galectin-1.

The data presented in this ms support the hypothesis that FGL2 blocks GM-CSF induced differentiation of CD103+ dendritic cells to initiate an anti-tumor CD8+ T cell response. In the absence of FGL2 the authors show that dendritic cell differentiation and stimulation of the anti-tumor immune response are not impaired.

One item that the authors would need to evaluate in further detail is any contribution made by NK cells. The strains of mice used, and the kinetics of tumor rejection (below 7 days), suggest at least a significant contribution of NK cells to tumor rejection. Biochemical and knockout experiments also do not discard a contribution of NK cells, as both CD8, and CD103 have been reported to be expressed in NK cells.

In summary, an original and interesting manuscript on novel mechanisms by which FGL2 regulates anti-tumor immune responses. The experimental results are sound and convincing. The mechanisms need to be confirmed in respect of a possible contribution of NK cells. Such experiments would not be too complex, and would provide further strengthening of the mechanistic hypothesis supporting the observed immune regulation of brain tumors.

Reviewer #3:

Remarks to the Author:

Manuscript#: NCOMMS-18-14621.

Title: Knockout immune checkpoint regulator FGL2 in tumor cells impairs tumor progression in the CNS by facilitating CD103+ dendritic cell differentiation

Summary: The authors present an elegant study demonstrating that Fibrinogen-like protein 2 (FGL2), an immune checkpoint regulator expressed in glioblastoma (GBM) limits the induction of antitumor T-cell immunity by impairing the accumulation of CD103+ dendritic cells by inhibiting GM-CSF signaling on differentiation DCs. GBM tumors lacking FGL2 allowed infiltration, differentiation and LN migration of CD103+ DCs. This differentiation and maturation process of CD103+DCs was required for the initiation of CD8+ T cell immune responses which ultimately were responsible for tumor rejection. These findings are relevant to GBM patients because low levels of FGL2 expression and high concurrent GM-CSF expression is associated with long-term survival.

Comments to the author

Major comments

None

Minor comments

- Please provide more explicit methods in the material and methods section as well as in the figure legends
- Please include the statistical tests utilized to determine significance in the figure legends
- Were the FGL2^{-/-} mice and CD8^{-/-} mice utilized in the experiments previously published? if so please include reference. If not, please describe briefly how these were generated
- The data presented in supplementary figure 5 (e-g) utilizing lymphocyte transfer from naïve C57BL/6 mixed with GL261 into NSG mice is quite a contrived design which limits the interpretation of the role of DCs in orchestrating T cell responses, which is the main focus of this paper.
- Please clarify the reasoning behind the utilizing subcutaneous implantations of GL261 and DBT tumors when evaluating the contribution of T cells in tumor regression, and the rejection of tumor rechallenge.
- Please indicate how western blot relative expression level was calculated and what statistical comparisons were performed to determine significance in the bar graphs.
- Please indicate in the materials and methods section how was FGL2^{low} versus FGL2^{high} determine. What was the cutoff and how was this reached? Same question regarding GM-CSF^{high}, CD8B^{high} and IFNG^{low} vs IFNG^{high}.

Reviewer #1:

1) In figure 4 there appear to be almost no DCs in the LNs and tumors. This is possibly due to gating out the CD11b⁺ subset, but also gating out many of the CD103⁺ DCs (which express moderate amounts of CD11b). Finally, there should be no CD8a DCs in peripheral tissues (unless TLNs are forming). Are these CD8a⁺ DCs the same population as the CD103⁺ DCs in this analysis? CD103⁺ DCs do not normally express CD8a. A full gating schema for LNs and tumors should be shown in the supplemental figures to demonstrate that the authors are able to accurately identify these populations.

Response and Revision: We changed the gating strategy to better define the CD103⁺ DC population. As suggested by this reviewer, the full gating strategy for TDLNs and tumors is included in Supplemental Fig. 4a and 4b. As shown in the revised Fig. 4a and Supplementary Fig. 4a-b, a unique subpopulation of CD103⁺/CD8a⁺ DCs —CD11b^{low/medium}MHCII⁺CD11c⁺Lin⁻CD103⁺/CD8a⁺—was clearly identified.

Several studies have shown that the distribution of CD8a⁺ DCs is restricted to lymphoid organs, but studies from others found infiltrated CD8a⁺ DCs in transplantable and spontaneous melanomas (Spranger, et al., 2015; Fuertes, et al., 2011; Preynat-Seauve, et al., 2006) and sarcomas (Berhanu, et al., 2006), which accounted for about 20% of infiltrated DCs at the tumor site. We used lineage staining (CD3⁻NK1.1⁻Gr1⁻Ter119⁻) to exclude any other possible cell types, yielding a small population of CD8a⁺ DCs (less than 10 per 10000 CD45⁺ cells) in tumor-bearing brain tissues (Fig. 4a). These data confirmed the presence of CD8a⁺ DCs in the tumor microenvironment. These CD8a⁺ DCs are not the same population as the CD103⁺ DCs.

2) Batf3-deficient mice are unable to mount an effective CD8 response, but this does not necessarily imply that the cells are being functionally regulated by the pathway of interest. For example, anti-PD-1 also fails in Batf3-deficient mice, even though DCs are not a target of this therapeutic approach. Indeed the authors show a direct effect of FGL2 on CD8⁺ T cells during CD3 stimulation in vitro. The OT-I adoptive transfer and FTY720 experiments imply a role for naïve T cell activation, but again do not discriminate between impacts on DCs or T cells.

Response and Revision: We agree that the Batf3-deficient mice experiment itself did not confirm that the target of FGL2 is DCs directly, but the following evidence (including new data)

strongly points to this conclusion. First, FGL2 suppresses CD103⁺ conventional DC differentiation in vivo (Fig. 4a); second, FGL2 suppresses bone marrow cell differentiation into CD103⁺ conventional DCs directly in vitro (Fig. 5a, new data); third, DCs induced by FGL2KO tumor in vivo or FGL2KO tumor-conditioned medium (FGL2KO-CM) in vitro showed greater antigen-presentation capacity than the DCs induced by Ctrl tumor or Ctrl tumor-conditioned medium (Ctrl-CM) (Fig. 4d and Fig. 5b, new data); fourth, DCs induced by implanting FGL2KO tumors showed greater in vivo antigen-presentation capacity than DCs derived from Ctrl-tumor bearing mice, as supported by more OT-I proliferation and CD8⁺ T cell priming (Fig. 4e and 4f); fifth, FGL2KO tumor progressed when DCs were depleted but regressed when DCs were restored in CD11c-DTR mice (Fig. 4b, new data); sixth, Batf3-dependent DCs were necessary for showing the effect of FGL2KO on tumor progression, as supported by the progression of FGL2KO tumors in Batf3-deficient mice (Fig. 4c). Collectively, these findings show that suppression of CD103⁺ DC differentiation by FGL2 is one major mechanism of the suppressive immune response in glioblastoma. There are also some lines of evidence from other labs showing that DCs are the target of FGL2: the proportion and absolute numbers of DCs (CD11c⁺MHCII⁺) were 30% higher in the spleens of FGL2^{-/-} mice than in those of FGL2^{+/+} mice (Itay Shalev, et al., 2008); a significantly greater number of DCs were obtained from in vitro cultures of bone marrow cells from FGL2^{-/-} mice than in FGL2^{+/+}-derived DC cultures (Itay Shalev, et al., 2008); and finally, rFGL2 suppressed LPS-induced bone marrow DC maturation (Itay Shalev, et al., 2008; Camie, et al. 2003).

3) There are no studies performed to indicate that the DCs treated with FGL2 are less functional, the slight reduction in CD103 expression notwithstanding. The data from Fig 5a suggests these differences may be due to artefactual gating out of activated cells (that are simply appearing B220 high) or altered maturation of plasmacytoid DCs.

Response and Revision: We agree with the reviewer's comment, and a new experiment was included in this revision to test the antigen-presenting function of DCs in the presence and absence of FGL2. In brief, DCs were cultured for 15 days with Ctrl-CM or FGL2-CM and then were loaded with soluble OVA protein and their antigen-presentation capacity tested. As expected, DCs cultured with FGL2KO-CM induced more OT-I T cell activation than those cultured with Ctrl-CM (Fig. 5b), indicating that bone marrow DCs treated with FGL2 were less functional. No significant difference in numbers of plasmacytoid DCs was observed between bone marrow cells cultured with Ctrl-CM and those cultured with FGL2-CM at day 15 (Supplementary Fig. 6b), indicating the specific impact of FGL2 on CD103⁺ DCs. In accordance with our data, Itay Shalev et al. showed that proportions of DC subsets (plasmacytoid DCs and monocytic DCs) in lymphoid tissues of FGL2^{-/-} mice were not changed, but the number of DCs was greater than in WT mice (Itay Shalev, et al., 2008).

4) Generation of CD103+ DCs from bone marrow has been reported to take about 15 days using a combination of GM-CSF and FLT-3L (See Mayer et al. Blood 2017). Little to no CD103 is

found on cells matured with either cytokine alone even after 7-10 days. It is therefore surprising that the authors are showing data using conditioned medium that induces high CD103 expression in 5 days. Additional phenotyping is required to claim that these are CD103+ DCs.

Response and Revision: CD103⁺ expression occurs before terminal differentiation of CD103⁺ DCs. FGL2 affects CD103 expression both before and after differentiated CD103⁺ DCs are detectable. As shown in revised Fig. 5a, the DCs treated with Ctrl-CM had a lower level of CD103 expression than the DCs treated with FGL2KO-CM on both days 5 and 15. Clec9A, a functional biomarker of CD103⁺ DCs, was expressed on conventional DCs on day 15 but not on day 5, suggesting that CD103 was induced early in the culture, but CD103⁺ conventional DCs were terminally differentiated at a later stage of the culture (i.e., day 15). The expression of CD103 by these cells is supported by others' findings that CD103 expression was detected on bone marrow cells cultured with FLT3L or FLT3L+GM-CSF during days 7~15 (Sathe, et al., 2011; Hope, et al., 2017; Yokota-Nakatsuma, et al., 2016; Mochizuki, et al., 2016). In our model, both GM-CSF and FLT3L were detected in Ctrl-CM and FGL2KO-CM via ELISA (Fig. 5c). CD103 expression induced on bone marrow cells by FGL2KO-CM on day 5 could be reversed by neutralizing the GM-CSF in the CM. These data show that CD103 expression can be induced by FGL2KO-CM and that the early expression of CD103 was partly induced by GM-CSF in the FGL2KO-CM. It also has been reported that NOTCH2 ligand Delta Like 1 (DL1), retinoic acid, bacterial products, and the tissue environment all contribute to the regulation of CD103 expression on DCs (Kirkling, et al., 2018; Balan, et al., 2018; Roe, et al., 2017). FLT3L+DL1 culture induced a high level of CD103 expression on bone marrow cells at day 7. Therefore, CD103 expression is not limited to day 15 only.

5) The data in Figure 1 suggests that endothelial cells and leukocytes may express a significant fraction of the FGL2 in the tumor. Although the functional data in the subsequent figures indicates an important role for FGL2 expression by tumors cells, the related text should be clear that the microenvironment is also a significant source. It would also be better to report the percentage of the subset positive for FGL2 for the flow cytometry data (e.g. 70% of CD31+ cells, not 1.8%).

Response and Revision: We agree with the reviewer and have revised the manuscript accordingly. As requested, we revised the description in the Results on page 5 as follows: "Portions of CD45⁺ immune cells (37.3±18.3% of CD45⁺ cells) and CD31⁺ endothelial cells (47.7±14.7% of CD31⁺ cells) were also FGL2 positive, but endothelial cells (1.0±0.6%) and leukocytes (5.7±3.0%) accounted for only small fractions of total cells from these tissues (Fig. 1c)". Therefore, the total number of FGL2⁺ endothelial cells and leukocytes was much smaller than the number of tumor cells. Notably, FGL2 knockout in mice had no impact on tumor progression in their brains (Fig. 2d, Supplementary Fig. 2e), further suggesting that FGL2 from host cells may play a minor role in GBM progression.

6) Fig S6D would benefit from a proliferation analysis.

Response and Revision: Proliferation analysis has been added to the new Supplementary Fig. 5d. We also placed the description in the context of the results on page 9 (Supplementary Fig. 5d) as “this secreted FGL2 suppressed T cell expansion (CFSE dilution), but not activation (CD25 expression) in vitro (Supplementary Fig. 5d)”.

7) *In Figure 7B/C the impact on survival is not very convincing. Focusing on the impact of FGL2 expression on CD8/IFNG (Fig 7d) would be more informative. However, the survival differences without FGL2 need to be shown, otherwise it is unclear if this adds anything to segregating out patients with high CD8/IFNG expression.*

Response and Revision: As suggested, the correlation assay between FGL2, CD8B, and IFNG expression, and the effect of single gene expression level on survival difference, are now provided in the new Fig. 7. FGL2 expression was significantly and negatively correlated with CD8B expression and positively correlated with IFNG expression.

We also provide the following statement in the context of the results on page 15 in Fig 7: “As expected, increased expression levels of GM-CSF (hazard ratio [HR]: 0.854, P=0.008) or CD8B (HR: 0.906, P=0.078) in GBM was associated with longer overall survival (OS) (Fig. 7a), while increased FGL2 expression was associated with poorer OS, though the difference was not statistically significant (HR: 1.113, P=0.071) (Fig. 7a). There is a statistically significant positive correlation between GM-CSF and CD8B expression (Fig. 7c) and a strong negative correlation between FGL2 and CD8B expression (Fig. 7d), suggesting that a low level of FGL2 and a high level of GM-CSF may be associated with high CD8⁺ T cell infiltration in tumors.”

8) *The discussion covers a wide range of topics, many of which are only tangentially related to the paper (e.g. B cells, DC vaccination).*

Response and revision: The material on B cells and DC vaccination has been removed from the revised Discussion.

9) *There is no data to indicate that FGL2 is an “immune checkpoint regulator”. These words should be removed from the title and text. A suggested title might be “FGL2 promotes glioblastoma progression by suppressing CD103⁺ dendritic cell differentiation” would be appropriate.*

Response and Revision: The statement “FGL2 is an immune checkpoint regulator” has been deleted. The title has been revised to “FGL2 promotes tumor progression in the CNS by suppressing CD103⁺ dendritic cell differentiation” as suggested.

Reviewer #2:

The manuscript by Yan et al. aims to provide evidence that FGL2 expressed by tumor cells reduces tumor growth by stimulating an immune response against tumors. They further suggest that this is achieved by FGL2 mediated differentiation of CD103+ dendritic cells and the stimulation of an anti-tumor T cell response.

Over the last years evidence has been presented that tumor cells encode a series of molecules that inhibit immune responses against the tumors themselves. These molecules do not exert their effects through modifying the activity of MDSCs or Tregs. A number of these molecules belong to the immune system, such as PD-L1, while others are unrelated to molecules known from the immune system. Tumors also express molecules that directly inhibit NK cells, such as LDH5 and galectin-1.

The data presented in this ms support the hypothesis that FGL2 blocks GM-CSF induced differentiation of CD103+ dendritic cells to initiate an anti-tumor CD8+ T cell response. In the absence of FGL2 the authors show that dendritic cell differentiation and stimulation of the anti-tumor immune response are not impaired.

One item that the authors would need to evaluate in further detail is any contribution made by NK cells. The strains of mice used, and the kinetics of tumor rejection (below 7 days), suggest at least a significant contribution of NK cells to tumor rejection. Biochemical and knockout experiments also do not discard a contribution of NK cells, as both CD8, and CD103 have been reported to be expressed in NK cells.

In summary, an original and interesting manuscript on novel mechanisms by which FGL2 regulates anti-tumor immune responses. The experimental results are sound and convincing. The mechanisms need to be confirmed in respect of a possible contribution of NK cells. Such experiments would not be too complex, and would provide further strengthening of the mechanistic hypothesis supporting the observed immune regulation of brain tumors.

Response and Revision: In the Results section, Fig. 3c (page 7), we state: “NK depletion had a modest negative effect on survival (P=0.14, Fig. 3c), depletion of CD8⁺ T cells completely reversed the survival benefit to mice implanted with GL261-FGL2KO cells (Fig. 3d; Supplementary Fig. 3)”. This line of evidence suggests a modest contribution of NK cells to tumor rejection, but not as strong as CD8⁺ T cells.

We also state that “The number of CD8⁺ T and CD4⁺ T cells was significantly greater in the brains of mice implanted with FGL2KO tumor cells than in those implanted with Ctrl tumor cells, but this was not the case with NK or NKT cells (Supplementary Fig. 4c)”. This suggests that the NK cells were not directly affected by FGL2 expression in tumor cells.

Furthermore, in revised Supplementary Fig. 4a and 4e, we show that very little CD8 and CD103 were expressed on NK cells from GL261-Ctrl or GL261-FGL2KO tumors, which confirmed that CD103 and CD8a on DCs, but not on NK cells, are regulated by FGL2.

Reviewer #3:

1) Please provide more explicit methods in the material and methods section as well as in the figure legends

Response and Revision: More explicit methods have been added in the revised figure Legends, Materials and Methods, and Supplementary Methods sections as suggested.

2) Please include the statistical tests utilized to determine significance in the figure legends

Response and Revision: The statistical tests utilized to determine significance have been added in the revised legend of each figure.

3) Were the FGL2^{-/-} mice and CD8^{-/-} mice utilized in the experiments previously published? if so please include reference. If not, please describe briefly how these were generated

Response and Revision: FGL2^{-/-} mice were a gift from Dr. Gary Levy (Toronto General Hospital / Research Institute, Toronto, ON, Canada) as we stated in the Materials and Methods. The published reference has been added.

CD8^{-/-} mice were generated in our laboratory and the methodology has not yet been published. A brief description has been added to the Materials and Methods as follows: “The RCAS/Ntv-a system was used to generate CD8^{-/-} mice^{3,4}. In brief, the Ntv-a transgene (avian cell surface receptor for subgroup A avian leukosis virus under the control of a glial progenitor-specific promoter derived from the human NES gene) and the CD8 α -targeted allele were moved from their respective genetic backgrounds onto the C57BL/6 background by marker-assisted backcrossing to yield Ntv-a/CD8 α ^{-/-} and Ntv-a/CD8 α ^{+/+} mice.”

4) The data presented in supplementary figure 5 (e-g) utilizing lymphocyte transfer from naïve C57BL/6 mixed with GL261 into NSG mice is quite a contrived design which limits the interpretation of the role of DCs in orchestrating T cell responses, which is the main focus of this paper.

Response and Revision: Thanks for pointing out this concern. To place our focus more clearly on the DC mechanism, we removed this Supplementary Figure 5 from the revised manuscript.

5) Please clarify the reasoning behind the utilizing subcutaneous implantations of GL261 and DBT tumors when evaluating the contribution of T cells in tumor regression, and the rejection of tumor rechallenge.

Response and Revision: Both intracranial and subcutaneous implantation models were used for testing brain tumor progression in mice in this study. The major reason for using a subcutaneous implantation model was to test the rejection of tumor rechallenge or induction of peripheral immune memory. To test the immune memory induction, we rechallenged tumor cells repeatedly (four times). We did not use repeated intracranial implantation because it causes too much suffering for the mice. We also tried to figure out whether the memory response induced by

FGL2KO tumor cells was systemic or local. Therefore, we chose a subcutaneous model for testing the rejection of the tumor rechallenge. Other experiments for testing the contribution of T cells in tumor regression by subcutaneous implantation were utilized to confirm that the effect of FGL2 on GL261 and DBT tumor progression was the same for the subcutaneous model as for the intracranial model. We deleted those data from the revised manuscript for the reason cited for the last question—namely to focus on the DC mechanism.

6) Please indicate how western blot relative expression level was calculated and what statistical comparisons were performed to determine significance in the bar graphs.

Response: In the Supplementary Methods, we have provided the following information: “ImageJ software was used to quantify blots. GAPDH was used as a control, and results were quantified by calculating the band intensity of protein X relative to GAPDH”. We have provided the method of statistical analysis in the legend of Fig. 5a: “Data were summarized as ratio changes from at least three independent experiments. The t-test was used to calculate the two-sided P values”.

7) Please indicate in the materials and methods section how was FGL2^{low} versus FGL2^{high} determine. What was the cutoff and how was this reached? Same question regarding GM-CSF^{high}, CD8B^{high} and IFNG^{low} vs IFNG^{high}.

Response and Revision: Thank you for pointing out this omission. We have clarified in the Materials and Methods section that the commonly used median gene expression z-score of the complete cohort (401 patients) was utilized as a cutoff value for classification of patients into high and low expression groups for each individual gene of interest, including FGL2 (cutoff value 0.33), GM-CSF (cutoff value -0.02), CD8B (cutoff value 0.01), and IFNG (cutoff value -0.24). According to the cutoff value of each gene, the patients with expression z-score values higher than the median were classified into the “high-expression” group (denoted as FGL2^{high}, GM-CSF^{high}, CD8B^{high}, and IFNG^{high}) and the patients with expression z-scores less than the median were classified into the “low-expression” group (denoted as FGL2^{low}, GM-CSF^{low}, CD8B^{low}, and IFNG^{low}). We also considered different groupings of patients with different combined expression levels based on multiple genes, for example, comparing patients in the group with combined “FGL2^{high} and CD8B^{low} and IFNG^{low}” with patients in the group with “FGL2^{low} and CD8B^{high} and IFNG^{high}”.

All changes in the manuscript are denoted by red text. We appreciate the reviewers’ comments and believe these changes have increased the accuracy and clarity of this manuscript.

Warm regards,

Shulin Li, PhD

Reviewers' Comments:

Reviewer #1:

Remarks to the Author:

The authors have addressed all of my concerns.

Reviewer #2:

Remarks to the Author:

The authors have satisfactorily addressed all my queries. The bar graphs in Figure 5 should be analyzed by ANOVA.

Reviewer #3:

Remarks to the Author:

Comments to Author:

The authors have satisfactorily addressed all prior concerns.